# A Partially Supervised Reinforcement Learning Framework for Visual Active Search

**Anindya Sarkar   Nathan Jacobs   Yevgeniy Vorobeychik**
{anindya, jacobsn, yvorobeychik}@wustl.edu,
Department of Computer Science and Engineering
Washington University in St. Louis

## Abstract

Visual active search (VAS) has been proposed as a modeling framework in which visual cues are used to guide exploration, with the goal of identifying regions of interest in a large geospatial area. Its potential applications include identifying hot spots of rare wildlife poaching activity, search-and-rescue scenarios, identifying illegal trafficking of weapons, drugs, or people, and many others. State of the art approaches to VAS include applications of deep reinforcement learning (DRL), which yield end-to-end search policies, and traditional active search, which combines predictions with custom algorithmic approaches. While the DRL framework has been shown to greatly outperform traditional active search in such domains, its end-to-end nature does not make full use of supervised information attained either during training, or during actual search, a significant limitation if search tasks differ significantly from those in the training distribution. We propose an approach that combines the strength of both DRL and conventional active search by decomposing the search policy into a prediction module, which produces a geospatial distribution of regions of interest based on task embedding and search history, and a search module, which takes the predictions and search history as input and outputs the search distribution. We develop a novel meta-learning approach for jointly learning the resulting combined policy that can make effective use of supervised information obtained both at training and decision time. Our extensive experiments demonstrate that the proposed representation and meta-learning frameworks significantly outperform state of the art in visual active search on several problem domains.

## 1   Introduction

Consider a scenario where a child is abducted and law enforcement needs to scan across hundreds of potential regions from a helicopter for a particular vehicle. An important strategy in such a search and rescue portfolio is to obtain aerial imagery using drones that helps detect a target object of interest (e.g., the abductor's car) [1, 2, 3, 4, 5]. The quality of the resulting photographs, however, is generally somewhat poor, making the detection problem extremely difficult. Moreover, security officers can only inspect relatively few small regions to confirm search and rescue activity, doing so sequentially.

We can distill some key generalizable structure from this scenario: given a broad area image (often with a relatively low resolution), sequentially query small areas within it (e.g., by sending security officers to the associated regions, on the ground) to identify as many target objects as possible. The number of queries we can make is typically limited, for example, by budget or resource constraints. Moreover, query results (e.g., detected search and rescue activity in a particular region) are *highly informative about the locations of target objects in other regions*, for example, due to spatial correlation. We refer to this general modeling framework as *visual active search (VAS)*. Numerous other scenarios share this broad structure, such as identification of drug or human trafficking sites, anti-poaching enforcement activities, identifying landmarks, and many others. Sarkar et al. [6]

37th Conference on Neural Information Processing Systems (NeurIPS 2023).

recently proposed a visual active search (VAS) framework for geospatial exploration, as well as a deep reinforcement learning (DRL) approach for learning a search policy. However, the efficacy of this DRL approach remains limited by its inability to adapt to search tasks that differ significantly from those in the training data, with the DRL approach struggling in such settings even when combined with test-time adaptation methods.

To gain intuition about the importance of domain adaptivity in VAS, consider Figure 1. Suppose, we pre-train a policy by leveraging a fully annotated search tasks with *large vehicle* as a target class to learn a search policy. Now the challenge in visual active search problem is how to utilize such a policy for our current task, that is to search for *small car* given a broad aerial image. As depicted in Figure 1, an adaptive policy (right) initially makes a mistake but then quickly adapts to the current task by efficiently leveraging information obtained in response to queries to learning from its mistakes. In contrast, a non-adaptive policy (left) keeps repeating its

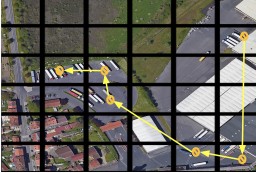 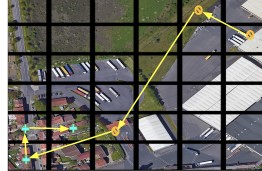

(a) Search strategy of a non-adaptive policy pre-trained on *large vehicle* as a target.

(b) Search strategy of an adaptive policy pre-trained on *large vehicle* as a target.

Figure 1: Comparative search strategy of non-adaptive and adaptive policy with *small car* as a target.

mistakes and ultimately fails to find the region containing the target object.

Indeed, traditional active search approaches have been designed precisely with such adaptivity in mind [7, 8, 9, 10] by combining an explicit machine learning model that predicts labels over inputs with custom algorithmic approaches aiming to balance exploration (which improves prediction efficacy) and exploitation (to identify as many actual objects of interest as possible within a limited budget). However, as Sarkar et al. [6] have shown, such approaches perform poorly in VAS settings compared to DRL, despite the lack of effective domain adaptivity of the latter.

We combine the best of traditional active search and the state-of-the-art DRL-based VAS approach by developing a partially-supervised reinforcement learning framework for VAS (PSVAS). A PSVAS policy architecture is comprised of two module: (i) A *task-specific prediction module* that learns to predict the locations of the target object based on the task aerial image and labels resulting from queries during the search process, and (ii) a *task-agnostic search module* that learns a search policy given the predictions provided by the prediction module and prior search results. The key advantage of this decomposition is that it enables us to use supervised information *observed at decision time* to update the parameters of the prediction module, without changing the search module. Furthermore, to learn search policies that are effective in the context of evolving predictions during the search, we propose a novel meta-learning approach to jointly learning the search module with the *initialization* parameters of the prediction module (used at the beginning of each task). Finally, we generalize the original VAS framework to allow for multiple simultaneous queries (a common setting in practice), and develop the PSVAS framework and a meta-learning approach for such settings.

In summary, we make the following contributions:

- A novel partially supervised reinforcement learning (PSVAS) framework that enables effective adaptation of VAS search policies to out-of-distribution search tasks.
- A novel meta-learning approach (MPS-VAS) to learn initialization parameters of the prediction module jointly with a search policy that is robust to the evolving predictions during a search task.
- A generalization of the VAS problem to domains in which we can make multiple simultaneous queries, and a variant of MPS-VAS that learns how to choose a subset of queries to make in each search iteration.
- An extensive experimental evaluation on two publicly available satellite imagery datasets, xView and DOTA, in a variety of unknown target settings, demonstrating that the proposed approaches significantly outperform all baselines. Our code is publicly available at this link.

## 2 Preliminaries

We consider a generalization of the *visual active search (VAS)* problem proposed by Sarkar et al. [6]. The basic building block of VAS is a *task*, which centers around an aerial image $x$ divided into

$N$ grid cells, so that $x = (x^{(1)}, x^{(2)}, ..., x^{(N)})$, with each grid cell a subimage. Broadly, the goal is to identify as many target objects of interest through iterative exploration of these grid cells as possible, subject to a budget constraint $\mathcal{C}$. To this end, we represent the subset of grids containing the target object by associating each grid cell $j$ with a binary label $y^{(j)} \in \{0, 1\}$, where $y^{(j)} = 1$ if the grid cell $j$ contains the target object, and 0 otherwise. The complete label vector associated with the task is $y = (y^{(1)}, y^{(2)}, ..., y^{(N)})$. When we are faced with the task at decision time, we have no direct information about $y$, but when we query a grid cell $j$, we obtain the ground truth label $y^{(j)}$ for this cell. Moreover, if $y^{(j)} = 1$, we also accrue utility from exploring $j$. In the original variant of VAS, we can make a single query $j$ in each time step. Here, we consider a natural generalization where we have $R$ query resources (for example, $R < N$ patrol units identifying traps in a wildlife conservation setting), so that we can make $R$ queries in each time step. We assume that $R$ is constant for convenience; it is straightforward to generalize our approach below if $R$ is time-varying.

Let $c(j, k)$ be the cost of querying grid cell $k$ if we start in grid cell $j$. For the very first query, we can define a dummy initial grid cell $d$, so that cost function $c(d, k)$ captures the initial query cost. Let $q_t^r$ denote the set of queries performed in step $t$ by a query resource $r$. Our ultimate goal is to solve the following optimization problem:

$$\max_{\{q_t^r\}} U(x; \{q_t^r\}) \equiv \sum_t y^{(q_t^r)}$$

$$\text{s.t.} : \sum_{t \geq 0} \sum_{r=1}^{R} c(q_{t-1}^r, q_t^r) \leq \mathcal{C}. \tag{1}$$

Finally, we assume that we possess a collection of tasks (in this case, aerial images) for which we have annotated whether each grid cell contains the target object or not. This collection, which we will refer to as $\mathcal{D} = \{(x_i, y_i)\}$, is comprised of images $x_i$ with corresponding grid cell labels $y_i$, where each $x_i$ is composed of $N$ elements $(x_i^{(1)}, x_i^{(2)}, \ldots, x_i^{(N)})$ representing the cells in the image, and each $y_i$ contains $N$ corresponding labels $(y_i^{(1)}, y_i^{(2)}, \ldots, y_i^{(N)})$.

The central technical goal in VAS is to learn an effective search policy that maximizes the total number of targets discovered, on average, for a sequence of tasks $x$ on which we have no prior direct experience, given a set of resources $R$, exploration cost function $c(j, k)$, and total exploration budget $\mathcal{C}$. Sarkar et al. [6] proposed a deep reinforcement learning (DRL) approach in which they learned a search policy $\pi(x, o, B)$ that outputs at each time step $t$ the grid we should explore at the next step $t + 1$, given the task $x$, remaining budget $B$, and information produced from the sequence of previous queries encoded into an observation vector $o$ with $o^{(j)} = 2y^{(j)} - 1$ if $j$ has been explored, and $o^{(j)} = 0$ otherwise. The reward function for this DRL approach is naturally captured by $R(x, o, j) = y^{(j)}$.

Note that active search (including VAS) is qualitatively distinct from active learning [11, 12, 13]: in the latter, the goal is solely to learn to predict well, so that the entire query process serves the goal of improving predictions. In active search, in contrast, we aim to learn a search policy that balances exploration (improving our ability to predict where target objects are) and exploitation (actually finding such objects) within a limited budget. Indeed, it has been shown that active learning approaches are not competitive in the VAS context [6].

## 3   Partially-Supervised Reinforcement Learning Approach for VAS

The DRL approach for VAS proposed by Sarkar et al. [6] is end-to-end, producing a policy that is only partly adaptive to observations $o$ made during exploration for each task. In particular, this end-to-end policy aims to capture the tension between exploration and exploitation fundamental in active search [7, 9], without explicitly representing the central aim of exploration, which is to improve our ability to *predict* which grid cells in fact contain the target object. In conventional active search, in contrast, this is directly represented by learning a prediction function $f(x^{(j)})$ that is updated each time a grid $j$ is explored during the search process, so that exploration directly impacts our ability to predict target locations, and thereby make better query choices in the future. While the end-to-end approach implicitly learns this, it would only do so effectively so long as tasks $x$ we face at prediction time are closely related to those used in training. However, it does not take advantage of the *supervised* information we obtain about which grid cells actually contain the target object, either at training or

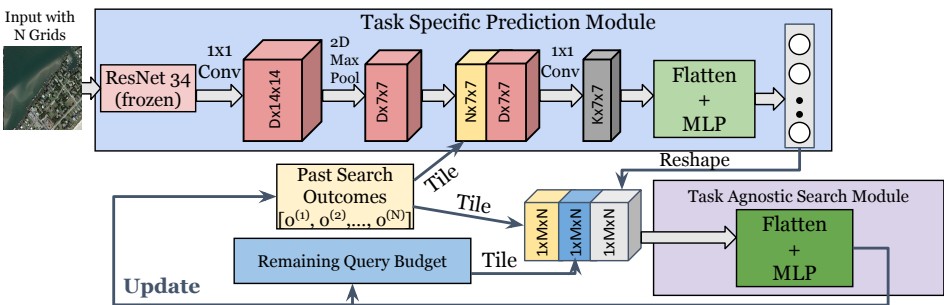

Figure 2: The PSVAS policy network architecture.

test time, potentially reducing the efficacy of learning as well as ability to adapt when task distribution changes.

To address this issue, we propose a novel *partially-supervised reinforcement learning* approach for visual active search, which we refer to as PSVAS. In PSVAS, the search policy we obtain is a composition of two modules: 1) the task-specific prediction module $f_\theta(x, o)$ and 2) the task-agnostic search module $g_\phi(p, o, B)$, where $\theta$ and $\phi$ are trainable parameters, where $p = f_\theta(x, o)$ is the vector of predicted probabilities with $p^{(j)}$ the predicted probability of a target in grid cell $j$ (see Figure 2, and Supplement Section A for further details about the policy network architecture). Conceptually, $f_\theta$ makes predictions based solely on the task $x$ and the prediction-relevant information gathered during the search $o$, while $g$ relies *solely* on the information relevant to the search itself: predicted locations of objects $p$, observations acquired during the search $o$, and remaining budget $B$. The search policy is then the composition of these modules, $\pi(x, o, B) = g_\phi(f_\theta(x, o), o, B)$.

Although in principle we can still train the policy $\pi$ above end-to-end (effectively collapsing the composition), a key advantage of PSVAS is that it enables us to directly make use of supervised information about true labels $y$ as they are observed either at training or decision time, using these to update $f_\theta$ in both cases. In prior work that relies solely on end-to-end policy representation, there was no straightforward way to take advantage of this information. Specifically, we train the policy using the following objective function:

$$\mathcal{L}_{\text{PARVS}} = \mathcal{L}_{RL} + \lambda \mathcal{L}_{BCE},$$

where $\lambda$ is a hyperparameter. We represent $\mathcal{L}_{RL}$ loss as follows:

$$\nabla \mathcal{L}_{RL} = -\sum_{i=1}^{M} \sum_{t=1}^{T_i} \mathbb{1}_{\sum_{t \geq 0} c(q_{t-1}, q_t) \leq \mathcal{C}} \nabla \log \pi_{(\theta, \phi)}(a_i^t | x_i, o_i^t, B_i^t) \mathbb{R}_i^t$$

Where $M$ is the number of example search task seen during training and $\mathbb{R}_i^t$ is the discounted cumulative reward defined as $\mathbb{R}_i^t = \sum_{k=t}^{T} \gamma^{k-t} R_i^k$ with a discount factor $\gamma \in [0, 1]$. Note that the RL loss in this approach *also updates the parameters of the prediction module*, $\theta$, in an end-to-end fashion. We also represents $\mathcal{L}_{BCE}$ as follows: $\mathcal{L}_{BCE} = \sum_{i=1}^{M} -(y_i \log(p_i) + (1 - y_i) \log(1 - p_i))$.

The PSVAS algorithm is more formally presented in Algorithm 1. The combined RL and supervised loss yields a balance between using supervised information to improve the quality of initial predictions at the beginning of the episode and ensuring that these serve the goal of producing the best episode-level policies. However, it is still crucial to note that $f_\theta$ serves solely an *instrumental* role in the process, with learning a search policy that *effectively adapts to each task* the primary goal. Consequently, we ultimately wish to jointly learn $g_\phi$ and $f_\theta$ in such a way that $f_\theta$ facilitates adaptive search as it is updated during a search task at decision time. To address this, we propose a meta-learning approach, MPS-VAS, that learns $g_\phi$ along with an initial parametrization $\theta_0$ of $f_\theta$ for each task. At the beginning of each task, then, $\theta$ is initialized to $\theta_0$, and then updated as labels are observed after each query.

At the high level, MPS-VAS trains over a series of *episodes*, where each episode corresponds to a fully labeled training task $(x_i, y_i)$ and budget constraint $\mathcal{C}$ (we vary the budget constraints during training). We begin an episode $i$ with the current prediction function parameters $\theta_i = \theta$ and search policy parameters $\phi$. We fix $g_\phi$ over the length of the episode, and use it to generate a sequence of queries (since the policy is stochastic, it naturally induces exploration). After observing the label $y^{(j)}$

---

**Algorithm 1** The PSVAS algorithm.

---

**Require:** A search task instance $(x_i, y_i)$; budget constraint $\mathcal{C}$; Prediction function ($f$) with current parameters $\theta_i$, i.e., $f_{\theta_i}$; Search policy ($g$) with current parameters $\phi_i$, i.e., $g_{\phi_i}$;
1: **Initialize** $o^0 = [0...0]$; $B^0 = \mathcal{C}$; step $t = 0$
2: **while** $B^t > 0$ **do**
3:     $\tilde{y} = f_{\theta_i}(x_i, o^t)$
4:     $j \leftarrow Sample_{j \in \{Unexplored\ Grids\}}[g_{\phi_i}(\tilde{y}, o^t, B^t)]$
5:     Query grid cell with index $j$ and observe true label $y^{(j)}$.
6:     Obtain reward $R^t = y^{(j)}$, Update $o^t$ to $o^{t+1}$ with $o^{(j)} = 2y^{(j)} - 1$, and update $B^t$ to $B^{t+1}$ with $B^{t+1} = B^t - c(k, j)$ (assuming we query $k$'th grid at $(t-1)$).
7:     Collect transition tuple ($\tau$) at step t, i.e., $\tau^t = ($ state $= (x_i, o^t, B^t)$, action $= j$, reward $= R^t$, next state $= (x_i, o^{t+1}, B^{t+1})$ ).
8:     $t \leftarrow t + 1$
9: **end while**
10: Update the prediction and search policy parameters, i.e., $\theta_i$ and $\phi_i$ using ($\mathcal{L}_{PSVAS}$) based on the collected transition tuples ($\tau^t$) and the collected labels ($y^{(j)}$) throughout the episode.
11: **Return** updated prediction and search policy parameters, i.e., $\theta_{i+1}$ and $\phi_{i+1}$ respectively.

---

for each queried grid cell $j$ during the episode, we update $f_{\theta_i}$ using a standard supervised (binary cross-entropy) loss. At the completion of the episode (once we have exhausted the search budget $\mathcal{C}$), we update the policy parameters, as well as the *initialization* prediction function parameters $\theta$ by combining RL and supervised loss. For the search module, we use the accumulated sum of rewards $R_i = \sum_j y^{(j)}$ over grids $j$ explored during the episode, with the RL loss $\mathcal{L}_{RL}$ based on the *REINFORCE* algorithm [14]. For the prediction module, we use the collected labels $y^{(j)}$ during the episode and the standard binary cross-entropy loss. Finally, the MPS-VAS loss also explicitly trades off the RL and supervised loss: $\mathcal{L}_{\text{MLPARVS}} = (\mathcal{L}_{RL} + \lambda \mathcal{L}_{BCE})$. *The proposed meta-learning approach thus explicitly trains the policy to account for the evolution of the prediction during the episode.* The full MPS-VAS is presented more formally in Algorithm 2.

---

**Algorithm 2** The MPS-VAS meta-learning algorithm.

---

**Require:** A search task instance $(x_i, y_i)$; budget constraint $\mathcal{C}$; Prediction function ($f$) with current parameters $\theta_i^0 = \theta_i$, i.e., $f_{\theta_i^0}$; Search policy ($g$) with current parameters $\phi_i$, i.e., $g_{\phi_i}$;
1: **Initialize** $o^0 = [0...0]$; $B^0 = \mathcal{C}$; step $t = 0$
2: **while** $B^t > 0$ **do**
3:     $\tilde{y} = f_{\theta_i^t}(x_i, o^t)$
4:     $j \leftarrow Sample_{j \in \{Unexplored\ Grids\}}[g_{\phi_i}(\tilde{y}, o^t, B^t)]$
5:     Query grid cell with index $j$ and observe true label $y^{(j)}$.
6:     Update $\theta_i^t$ to $\theta_i^{t+1}$ using $\mathcal{L}_{BCE}$ loss between $\tilde{y}$ and pseudo label $\hat{y}$, defined as
        $\hat{y} \leftarrow \begin{cases} y^{(j)} & \text{if } y^{(j)} \text{ is Observed} \\ \tilde{y}^{(j)} & \text{if } y^{(j)} \text{ is Unobserved.} \end{cases}$
7:     Obtain reward $R^t = y^{(j)}$, Update $o^t$ to $o^{t+1}$ with $o^{(j)} = 2y^{(j)} - 1$, and update $B^t$ to $B^{t+1}$ with $B^{t+1} = B^t - c(k, j)$ (assuming we query $k$'th grid at $(t-1)$).
8:     Collect transition tuple $\tau$ at step t, i.e., $\tau^t = ($ state $= (x_i, o^t, B^t)$, action $= j$, reward $= R^t$, next state $= (x_i, o^{t+1}, B^{t+1})$ ).
9:     $t \leftarrow t + 1$
10: **end while**
11: Update search policy parameters $\phi_i$ using ($\mathcal{L}_{RL}$) based on the collected transition tuples ($\tau^t$) throughout the episode and update initial prediction function parameters $\theta_i$ using ($\mathcal{L}_{BCE}$) based on the collected labels ($y^{(j)}$) throughout the episode.
12: **Return** updated prediction and search policy parameters, i.e., $\theta_{i+1}$ and $\phi_{i+1}$ respectively.

---

The search policy $\pi$ produces a probability distribution over grid cells. However, since in our setting no advantage can be gained by querying previously queried grid cells, we simply renormalize the probability distribution induced by $\pi$ over the remaining grid cells, both during training and at decision time. Note that during inference, in both PSVAS and MPS-VAS framework, we freeze the parameters of the search module ($\phi$) and update the parameters of the prediction module ($\theta$) after

observing query outcomes at each step using $\mathcal{L}_{BCE}$ loss between predicted label and pseudo label as shown in step (6) of Algorithm 2.

The discussion thus far effectively assumed that $R = 1$, that is, we make only a single query in each search time step. Next, we describe the generalization of our approach when we have multiple query resources $R > 1$. First, note that the prediction module $f_\theta$ is unaffected, since the number of query resources only pertain to the actual search strategy $g_\phi$. One way to handle $R$ queries is to simply sample the search module (which is stochastic) iteratively $R$ times without replacement during training, and to greedily choose the most probable $R$ grid cells to query at decision time. We refer to this as MPS-VAS-TOPK. However, since the underlying problem is now combinatorial (the choice of $R$ queries out of $N$ grid cells), such a greedy policy may fail to capture important interdependencies among such search decisions.

To address this, we propose a novel policy architecture which is designed to learn how to optimize such a heuristic greedy approach for combinatorial grid cell selection in a way that is non-myopic and accounts for the interdependent effects of sequential greedy choices. Specifically, let $\psi$ be a vector corresponding to the grid cells, with $\psi_j = 1$ if grid cell $j$ has either been queried in the past (during previous search steps), or has been already chosen to be queried in the current search step, and $\psi_j = 0$ otherwise. Thus, $\psi$ encodes the choices that have already been made, and enables the policy to learn the best next sequential choice to make using a greedy procedure through the same RL framework that we described above. We refer to this approach as MPS-VAS-MQ (in reference to multiple queries).

## 4 Experiments

### 4.1 Experiment Setup

Since the goal of active search is to maximize the number of target objects identified, we use *average number of targets* identified through exploration (**ANT**) as our evaluation metric.

We consider two ways of generating query costs: (i) $c(i, j) = 1$ for all $i, j$, where $\mathcal{C}$ is just the number of queries, and (ii) $c(i, j)$ is based on Manhattan distance between $i$ and $j$. Most of the results we present in the main paper reflect the Manhattan distance based setting; we also report the results for uniform query costs in the Supplement. We use $\lambda = 0.1$ in all the experiments. We present the details of policy architecture and hyper-parameter details for each different experimental settings in the Supplement.

**Baselines** We compare the proposed PSVAS and MPS-VAS policy learning framework to the following baselines.

- *Random Search (RS)*, in which unexplored grid cells are selected uniformly at random.
- *Conventional Active Search (AS)* proposed by Jiang et. al. [9], using a low-dimensional feature representation for each image grid from the same feature extraction network as in our approach.
- *Greedy Classification (GC)*, in which we train a classifier $\psi_{GC}$ to determine whether a grid contains a target object. We then prioritize the search in grids with the highest probability of containing the target object until the search budget is depleted.
- *Active Learning (AL)*, in which the first grid is selected randomly for querying, and the remaining grids are chosen using a state-of-the-art active learning approach by Yoo et al. [13] until the search budget is saturated.
- *Greedy Selection (GS)*, proposed by Uzkent et al. [15], that trains a policy $\phi_{GS}$ to assign a probability of zooming into each grid cell $j$. We use this policy to select grids greedily until the search budget $\mathcal{C}$ is exhausted.
- *End-to-end visual active search (E2EVAS)*, the state-of-the-art approach for VAS proposed by Sarkar et al. [6].

When dealing with a multi-query scenario, we compare the effectiveness of MPS-VAS-TOPK and MPS-VAS-MQ.

**Datasets** We evaluate the proposed approach using two datasets: xView [16] and DOTA [17]. xView is a satellite imagery dataset which consists of large satellite images representing 60 categories, with approximately 3000 pixels in each dimensions. We use 67% and 33% of the large satellite

Table 1: **ANT** comparisons when trained with *small car* as target on xView in single-query setting.

| | Test with Helicopter as Target | | | Test with SB as Target | | | Test with Building as Target | | |
|---|---|---|---|---|---|---|---|---|---|
| Method | $\mathcal{C}$ = 25 | $\mathcal{C}$ = 50 | $\mathcal{C}$ = 75 | $\mathcal{C}$ = 25 | $\mathcal{C}$ = 50 | $\mathcal{C}$ = 75 | $\mathcal{C}$ = 25 | $\mathcal{C}$ = 50 | $\mathcal{C}$ = 75 |
| RS | 0.16 | 0.39 | 0.72 | 0.35 | 0.71 | 1.02 | 2.41 | 3.56 | 4.62 |
| GC | 0.29 | 0.55 | 0.93 | 0.52 | 0.95 | 1.21 | 3.94 | 5.14 | 6.61 |
| GS [15] | 0.41 | 0.68 | 1.08 | 0.61 | 1.03 | 1.26 | 4.51 | 5.80 | 6.82 |
| AL [13] | 0.27 | 0.54 | 0.92 | 0.52 | 0.93 | 1.18 | 3.92 | 5.12 | 6.60 |
| AS [9] | 0.25 | 0.46 | 0.83 | 0.51 | 0.95 | 1.20 | 3.79 | 5.01 | 6.34 |
| E2EVAS [6] | 0.53 | 0.83 | 1.25 | 0.67 | 1.10 | 1.30 | 5.85 | 9.26 | 11.96 |
| OnlineTTA[6] | 0.54 | 0.84 | 1.26 | 0.68 | 1.10 | 1.32 | 5.86 | 9.26 | 11.98 |
| **PSVAS** | **0.87** | **1.08** | **1.28** | **0.93** | **1.23** | **1.66** | **6.81** | **10.53** | **13.44** |
| **MPS-VAS** | **0.92** | **1.13** | **1.38** | **1.07** | **1.67** | **2.10** | **6.83** | **10.59** | **13.64** |
| | Test with CC as Target | | | Test with SC as Target | | | Test with Helipad as Target | | |
| Method | $\mathcal{C}$ = 25 | $\mathcal{C}$ = 50 | $\mathcal{C}$ = 75 | $\mathcal{C}$ = 25 | $\mathcal{C}$ = 50 | $\mathcal{C}$ = 75 | $\mathcal{C}$ = 25 | $\mathcal{C}$ = 50 | $\mathcal{C}$ = 75 |
| RS | 0.57 | 1.18 | 1.72 | 1.80 | 3.40 | 5.11 | 0.19 | 0.42 | 0.72 |
| GC | 1.05 | 1.81 | 2.14 | 2.64 | 4.88 | 7.04 | 0.41 | 0.78 | 1.04 |
| GS [15] | 1.12 | 1.97 | 2.48 | 3.35 | 5.39 | 7.52 | 0.54 | 0.93 | 1.12 |
| AL [13] | 1.04 | 1.77 | 2.12 | 2.62 | 4.88 | 7.03 | 0.40 | 0.76 | 1.03 |
| AS [9] | 1.02 | 1.61 | 2.03 | 2.43 | 4.61 | 6.95 | 0.38 | 0.75 | 0.98 |
| E2EVAS [6] | 1.43 | 2.31 | 2.98 | 4.73 | 7.43 | 9.59 | 0.81 | 1.20 | 1.46 |
| OnlineTTA[6] | 1.43 | 2.33 | 2.99 | 4.75 | 7.44 | 9.59 | 0.83 | 1.21 | 1.46 |
| **PSVAS** | **1.62** | **2.49** | **3.14** | **5.51** | **8.33** | **10.52** | **0.91** | **1.22** | **1.47** |
| **MPS-VAS** | **1.74** | **2.64** | **3.47** | **5.55** | **8.40** | **10.69** | **0.96** | **1.30** | **1.63** |

images to train and test the policy network respectively. DOTA is also a satellite imagery dataset. We rescale the original $3000 \times 3000px$ images to $1200 \times 1200px$.

## 4.2 Single Query Setting

We begin by considering a setting with a single query resource, as in most prior work. We first evaluate the proposed method on the xView dataset with varying search budget $\mathcal{C} \in \{25, 50, 75\}$ and the number of equal sized grid cells $N = 49$. We train the policy with *small car* as target and evaluate the performance of the policy with the following target classes : *Small Car* (SC), *Helicopter*, *Sail Boat* (SB), *Construction Cite* (CC), *Building*, and *Helipad*. As the dataset contains variable size images, we take random crops of $3500 \times 3500$ for $N = 49$, ensuring equal grid cell sizes. We present the results with different values of $N$ in the Supplement. The results with $N = 49$ are presented in Table 1. We observe significant improvements in performance of the proposed PSVAS approach compared to all baselines in each different target setting, ranging from 3 to 25% improvement relative to the most competitive E2EVAS method. The significance of utilizing supervised information of true labels $y$, which are observed after each query at inference time, is supported by the obtained results. This highlights the effectiveness of the PSVAS framework, which enables us to update task-specific prediction module $f$ by leveraging such crucial information in an efficient manner. The experimental outcomes also indicates two consistent trends. In each target setting, the overall search performance improves as $\mathcal{C}$ increases, and the relative advantage of MPS-VAS over PSVAS increases, as it is better able to exploit the observed outcomes and in turn improve the policy further for the larger search budget $\mathcal{C}$. We also observe that the extent of improvement in performance is greater when there is a greater difference between the target class used in training and the one used during inference. For example, when the target class is a *sail boat*, the improvement in performance of MPS-VAS in comparison to PSVAS ranges between 15% to 35%. However, if the target class is a *small car*, the improvement in performance of MPS-VAS is only between 1% to 2%. Considering all different target settings, performance improvement of MPS-VAS in comparison to PSVAS, ranges from 1% to 35%. The results demonstrate the efficacy of the MPS-VAS framework in learning a search policy that enables adaptive search. Figure 3 demonstrates the exploration strategies of different policies that are trained on small car as the target class and test on sail boat as the target. The figure showcases the different exploration behaviors exhibited by each policy in response to the target class, highlighting the impact of the proposed *adaptive search* framework on the resulting exploration strategies. In Table 2, we present similar results on the DOTA dataset with $N$ as 64. Here, we train the policy with *Large Vehicle* as target and evaluate the policy with the following target classes: *Ship*, *Large Vehicle* (LV), *Harbor*, *Helicopter*, *Plane*, and *Roundabout* (RB). We observe that MPS-VAS approach significantly outperforms all other baseline methods across different target scenarios. This shows the importance of MPS-VAS framework for deploying visual active search when search tasks differ significantly from those that are used for training.

Table 2: **ANT** comparisons when trained with *large vehicle* as target on DOTA in single-query setting.

| | Test with Ship as Target | | | Test with LV as Target | | | Test with Harbor as Target | | |
|---|---|---|---|---|---|---|---|---|---|
| Method | $\mathcal{C} = 25$ | $\mathcal{C} = 50$ | $\mathcal{C} = 75$ | $\mathcal{C} = 25$ | $\mathcal{C} = 50$ | $\mathcal{C} = 75$ | $\mathcal{C} = 25$ | $\mathcal{C} = 50$ | $\mathcal{C} = 75$ |
| Random | 1.08 | 2.11 | 3.06 | 1.48 | 2.96 | 3.91 | 1.41 | 2.72 | 3.90 |
| GC | 1.52 | 3.04 | 4.19 | 2.59 | 3.77 | 5.48 | 1.90 | 3.77 | 5.02 |
| GS[15] | 1.79 | 3.56 | 4.47 | 2.72 | 4.10 | 5.77 | 2.31 | 4.14 | 5.87 |
| AL[13] | 1.51 | 3.02 | 4.18 | 2.57 | 3.74 | 5.47 | 1.89 | 3.74 | 5.01 |
| AS[9] | 1.43 | 2.87 | 4.01 | 1.64 | 3.15 | 4.23 | 1.73 | 3.45 | 4.68 |
| VAS[6] | 2.45 | 4.37 | 5.87 | 5.33 | 8.47 | 10.51 | 3.12 | 5.04 | 6.82 |
| OnlineTTA[6] | 2.46 | 4.38 | 5.89 | 5.33 | 8.47 | 10.52 | 3.12 | 5.06 | 6.83 |
| **PSVAS** | **2.46** | **4.41** | **6.00** | **5.33** | **8.52** | **10.59** | **3.15** | **5.24** | **6.94** |
| **MPS-VAS** | **3.08** | **5.25** | **7.13** | **5.34** | **8.53** | **10.63** | **3.82** | **6.77** | **9.00** |

| | Test with Helicopter as Target | | | Test with Plane as Target | | | Test with Roundabout as Target | | |
|---|---|---|---|---|---|---|---|---|---|
| Method | $\mathcal{C} = 25$ | $\mathcal{C} = 50$ | $\mathcal{C} = 75$ | $\mathcal{C} = 25$ | $\mathcal{C} = 50$ | $\mathcal{C} = 75$ | $\mathcal{C} = 25$ | $\mathcal{C} = 50$ | $\mathcal{C} = 75$ |
| Random | 0.27 | 0.46 | 0.71 | 1.37 | 2.55 | 3.62 | 1.47 | 2.53 | 3.81 |
| GC | 0.38 | 0.59 | 0.91 | 1.96 | 3.14 | 4.02 | 1.62 | 2.86 | 4.23 |
| GS[15] | 0.42 | 0.66 | 1.01 | 2.26 | 3.56 | 4.71 | 1.91 | 3.24 | 4.68 |
| AL[13] | 0.37 | 0.58 | 0.89 | 1.94 | 3.14 | 3.99 | 1.61 | 2.82 | 4.21 |
| AS[9] | 0.34 | 0.53 | 0.82 | 1.89 | 3.06 | 3.92 | 1.55 | 2.71 | 4.06 |
| E2EVAS[6] | 0.47 | 0.72 | 1.05 | 3.07 | 4.87 | 6.34 | 3.05 | 4.94 | 6.34 |
| OnlineTTA[6] | 0.48 | 0.72 | 1.06 | 3.08 | 4.89 | 6.34 | 3.05 | 4.95 | 6.36 |
| **PSVAS** | **0.50** | **0.73** | **1.10** | **3.09** | **4.96** | **6.38** | **3.09** | **4.96** | **6.39** |
| **MPS-VAS** | **0.63** | **1.03** | **1.60** | **3.46** | **5.57** | **7.71** | **3.46** | **5.57** | **7.71** |

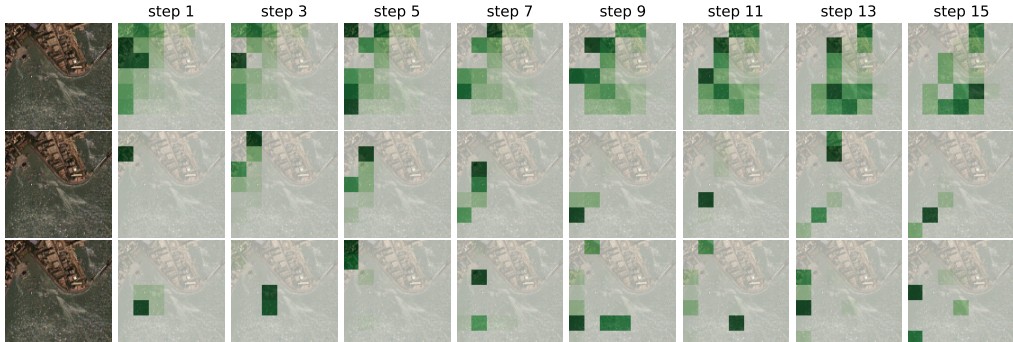

Figure 3: Query sequences, and corresponding heat maps (darker indicates higher probability), obtained using E2EVAS (top row), PSVAS (middle row), and MPS-VAS (bottom row).

## 4.3 Multi Query Setting

Next, we evaluate the proposed MPS-VAS-MQ approach on the xView and DOTA dataset in multi query setting. In Table 3, we present the results with varying search budget $\mathcal{C} = \{25, 50, 75\}$ and the number of equal sized grid cells $N = 49$. We consider $K = 3$ in all the experiments we perform in different target settings. We observe two general consistent trends across different target settings. First, the search performance of MPS-VAS in single query setting is always better than the performance of MPS-VAS-MQ in multi query settings, due to the fact that in single query setting the task-specific prediction module is updated $K$ times more frequently than in the multi-query setting. Second, across various target settings, the performance improvement of MPS-VAS-MQ over MPS-VAS-TOPK ranges from $0.08\%$ to $3.5\%$. In Table 4, we present similar result with the number of grid cells $N = 64$ and train the policy with *large vehicle* as the target class. We report the results with different values of $N$ in the Supplement. Here the improvement of MPS-VAS-MQ over MPS-VAS-TOPK is up to $3.5\%$ across different target settings, suggesting that there is added value from learning to capture interdependence in greedy search decisions.

## 4.4 Effect of $\lambda$ on Search Performance

We perform experiments with different choices of $\lambda$ and found $\lambda = 0.1$ to be the best choice across all different experimental setup. For comparison, here we report the results in the case when we train the policy with different values of $\lambda$ using *small car* as a target class and test the policy with

Table 3: **ANT** comparisons when trained with *small car* as target on xView in multi-query setting.

| | Test with Helicopter as Target | | | Test with SB as Target | | | Test with Building as Target | | |
|---|---|---|---|---|---|---|---|---|---|
| Method | $\mathcal{C} = 25$ | $\mathcal{C} = 50$ | $\mathcal{C} = 75$ | $\mathcal{C} = 25$ | $\mathcal{C} = 50$ | $\mathcal{C} = 75$ | $\mathcal{C} = 25$ | $\mathcal{C} = 50$ | $\mathcal{C} = 75$ |
| MPS-VAS-TOPK | 0.55 | 0.75 | 1.12 | 0.97 | 1.38 | 1.93 | 6.68 | 10.35 | 13.30 |
| **MPS-VAS-MQ** | **0.63** | **0.83** | **1.17** | **0.97** | **1.40** | **1.97** | **6.82** | **10.48** | **13.31** |

| | Test with CC as Target | | | Test with SC as Target | | | Test with Helipad as Target | | |
|---|---|---|---|---|---|---|---|---|---|
| Method | $\mathcal{C} = 25$ | $\mathcal{C} = 50$ | $\mathcal{C} = 75$ | $\mathcal{C} = 25$ | $\mathcal{C} = 50$ | $\mathcal{C} = 75$ | $\mathcal{C} = 25$ | $\mathcal{C} = 50$ | $\mathcal{C} = 75$ |
| MPS-VAS-TOPK | 1.63 | 2.59 | 3.36 | 5.41 | 8.28 | 10.30 | 0.81 | 1.07 | 1.23 |
| **MPS-VAS-MQ** | **1.67** | **2.60** | **3.38** | **5.47** | **8.34** | **10.38** | **0.84** | **1.14** | **1.30** |

Table 4: **ANT** comparisons when trained with *large vehicle* as target on DOTA in multi-query setting.

| | Test with Ship as Target | | | Test with LV as Target | | | Test with Harbor as Target | | |
|---|---|---|---|---|---|---|---|---|---|
| Method | $\mathcal{C} = 25$ | $\mathcal{C} = 50$ | $\mathcal{C} = 75$ | $\mathcal{C} = 25$ | $\mathcal{C} = 50$ | $\mathcal{C} = 75$ | $\mathcal{C} = 25$ | $\mathcal{C} = 50$ | $\mathcal{C} = 75$ |
| MPS-VAS-TOPK | 3.03 | 5.14 | 6.83 | 5.32 | 8.46 | 10.56 | 3.69 | 6.49 | 8.71 |
| **MPS-VAS-MQ** | **3.05** | **5.18** | **6.88** | **5.33** | **8.50** | **10.61** | **3.72** | **6.59** | **8.75** |

| | Test with Helicopter as Target | | | Test with Plane as Target | | | Test with RB as Target | | |
|---|---|---|---|---|---|---|---|---|---|
| Method | $\mathcal{C} = 25$ | $\mathcal{C} = 50$ | $\mathcal{C} = 75$ | $\mathcal{C} = 25$ | $\mathcal{C} = 50$ | $\mathcal{C} = 75$ | $\mathcal{C} = 25$ | $\mathcal{C} = 50$ | $\mathcal{C} = 75$ |
| MPS-VAS-TOPK | 0.57 | 0.96 | 1.39 | 3.38 | 5.43 | 7.64 | 3.29 | 5.43 | 7.62 |
| **MPS-VAS-MQ** | **0.60** | **1.01** | **1.46** | **3.41** | **5.48** | **7.66** | **3.38** | **5.50** | **7.66** |

*small car*, *building*, and *sail boat* as target on xView. We evaluate the policy with varying search budgets $\mathcal{C} \in \{25, 50, 75\}$ and the number of equal sized grid cells $N = 49$. In table 5, we provide the result for the PSVAS framework with small car as target. In table 6, we provide similar result for the MPS-VAS framework. Our empirical findings across all the experimental settings are quite consistent, and justify the choice of $\lambda = 0.1$.

Table 5: **ANT** comparisons when trained with *small car* as target for different values of $\lambda$ using the **PSVAS** Framework.

| | Test with Small Car as Target | | | Test with Building as Target | | | Test with Sail Boat as Target | | |
|---|---|---|---|---|---|---|---|---|---|
| $\lambda$ | $\mathcal{C} = 25$ | $\mathcal{C} = 50$ | $\mathcal{C} = 75$ | $\mathcal{C} = 25$ | $\mathcal{C} = 50$ | $\mathcal{C} = 75$ | $\mathcal{C} = 25$ | $\mathcal{C} = 50$ | $\mathcal{C} = 75$ |
| 0.001 | 4.96 | 7.75 | 9.74 | 6.08 | 9.64 | 12.35 | 0.74 | 1.12 | 1.43 |
| 0.01 | 5.02 | 7.87 | 9.96 | 6.37 | 9.95 | 12.77 | 0.88 | 1.19 | 1.54 |
| **0.1** | **5.51** | **8.33** | **10.52** | **6.81** | **10.53** | **13.44** | **0.93** | **1.23** | **1.66** |
| 1.0 | 5.10 | 7.98 | 10.04 | 6.39 | 10.16 | 12.81 | 0.89 | 1.20 | 1.59 |

Table 6: **ANT** comparisons when trained with *small car* as target for different values of $\lambda$ using the **MPS-VAS** Framework.

| | Test with Small Car as Target | | | Test with Building as Target | | | Test with Sail Boat as Target | | |
|---|---|---|---|---|---|---|---|---|---|
| $\lambda$ | $\mathcal{C} = 25$ | $\mathcal{C} = 50$ | $\mathcal{C} = 75$ | $\mathcal{C} = 25$ | $\mathcal{C} = 50$ | $\mathcal{C} = 75$ | $\mathcal{C} = 25$ | $\mathcal{C} = 50$ | $\mathcal{C} = 75$ |
| 0.001 | 4.99 | 7.82 | 9.90 | 6.15 | 9.74 | 12.44 | 0.83 | 1.22 | 1.53 |
| 0.01 | 5.06 | 7.93 | 10.03 | 6.41 | 10.09 | 12.89 | 0.98 | 1.46 | 1.87 |
| **0.1** | **5.55** | **8.40** | **10.69** | **6.83** | **10.59** | **13.64** | **1.07** | **1.67** | **2.10** |
| 1.0 | 5.12 | 8.01 | 10.12 | 6.46 | 10.21 | 12.96 | 1.01 | 1.52 | 1.90 |

## 4.5 Effectiveness of Task Specific Prediction Module on Search Performance

We analyse the importance of the task-specific prediction module in Sections *D.1* and *D.2* of Supplementary Material by freezing the prediction module parameters during inference time. Here, we additionally analyse the efficacy of the task-specific prediction module by setting $\lambda = 0$ while training the policy. We call the resulting policy *USVAS* (Un-Supervised VAS). We observe a significant drop in performance across all settings, demonstrating the importance of the Supervised prediction module in order to learn an effective search policy. Specifically, in the following table 7, we present the results when the policy is trained with small car on xView as a target, while the performance of the policy is evaluated for the following target classes: Small Car (SC), Helicopter, SailBoat (SB), Construction Site (CS), Building, and Helipad. We evaluate the policy with varying search budgets $\mathcal{C} \in \{25, 50, 75\}$ and the number of equal sized grid cells $N = 49$.

Table 7: **ANT** comparisons when trained with *small car* as target on xView.

| | Test with Small Car as Target | | | Test with Building as Target | | | Test with Sail Boat as Target | | |
|---|---|---|---|---|---|---|---|---|---|
| Method | $\mathcal{C} = 25$ | $\mathcal{C} = 50$ | $\mathcal{C} = 75$ | $\mathcal{C} = 25$ | $\mathcal{C} = 50$ | $\mathcal{C} = 75$ | $\mathcal{C} = 25$ | $\mathcal{C} = 50$ | $\mathcal{C} = 75$ |
| USVAS | 4.77 | 7.46 | 9.61 | 5.86 | 9.37 | 12.05 | 0.64 | 1.08 | 1.27 |
| PSVAS | 5.51 | 8.33 | 10.52 | 6.81 | 10.53 | 13.44 | 0.93 | 1.23 | 1.66 |
| MPS-VAS | **5.55** | **8.40** | **10.69** | **6.83** | **10.59** | **13.64** | **1.07** | **1.67** | **2.10** |
| | Test with Helicopter as Target | | | Test with CS as Target | | | Test with Helipad as Target | | |
| Method | $\mathcal{C} = 25$ | $\mathcal{C} = 50$ | $\mathcal{C} = 75$ | $\mathcal{C} = 25$ | $\mathcal{C} = 50$ | $\mathcal{C} = 75$ | $\mathcal{C} = 25$ | $\mathcal{C} = 50$ | $\mathcal{C} = 75$ |
| USVAS | 0.53 | 0.84 | 1.19 | 1.44 | 2.27 | 2.99 | 0.80 | 1.16 | 1.42 |
| PSVAS | 0.87 | 1.08 | 1.28 | 1.62 | 2.49 | 3.14 | 0.91 | 1.22 | 1.47 |
| MPS-VAS | **0.92** | **1.13** | **1.38** | **1.74** | **2.64** | **3.47** | **0.96** | **1.30** | **1.63** |

## 5 Related Work

**RL for Visual Navigation** RL has found broad applicability in visual navigation tasks [18, 19, 20, 21]. While these tasks share some similarities at a high level, such as requiring a sequence of visual navigation steps based on a local view of the environment, they often do not involve search budget constraints and rely on a predetermined kinematic model of motion. In contrast, our approach involves observing the full environment, albeit potentially at a lower resolution, and sequentially determining which regions to query without being constrained to a particular kinematic model. This highlights the distinctive nature of our approach compared to traditional visual navigation tasks and demonstrates the potential value of active search strategies in addressing budget-constrained settings.

**Active Search** Active Search was first introduced by Garnett et al. [7] as a means to discover members of valuable and rare classes rather than solely focusing on learning an accurate model as in Active Learning [11]. Subsequently, Jiang et al. [9, 22] proposed efficient nonmyopic active search techniques and incorporated search cost into the problem. Sarkar et al. [6] demonstrate that prior active search techniques do not scale well in high-dimensional visual space, and instead propose a DRL based visual active search framework for geo-spatial broad area search. However, the efficacy of the proposed approach by Sarkar et al. [6] can be limited when the search task varies between training and testing, which is often the case in real-world applications. In this work, we propose a novel framework that enables efficient and adaptive search in any previously unseen search task.

**Meta Learning** The concept of meta-learning, has consistently attracted attention in the field of machine learning [23, 24, 25, 26]. Finn et al. [25] present Model Agnostic Meta-Learning, a technique that utilizes SGD updates to rapidly adapt to new tasks. This approach, based on gradient-based meta-learning, can be seen as learning an effective parameter initialization, enabling the network to achieve good performance with just a few gradient updates. Wortsman et al. [27] introduces a self-adaptive visual navigation approach, which has the ability to learn and adapt to novel environments without the need for explicit supervision. Our work is significantly different than all these prior works as we observe true label at each step during search and hence the main challenge is how to leverage the supervised information in order to learn an efficient adaptive search policy.

**Foveated Processing of Large Images** Several studies have investigated the utilization of low-resolution images for guiding the selection of high-resolution regions to process [28, 29, 30, 31, 32, 33, 34, 35], with some employing reinforcement learning techniques [36, 15] to improve this process. However, our work differs significantly, as we focus on selecting a sequence of regions to query, where each query provides the true label, instead of a higher resolution image region. These labels are crucial for guiding further search and serving as an ultimate objective. As such, our approach tackles a unique challenge that differs from existing methods that rely on low-resolution imagery.

## 6 Conclusions

We present a novel approach for visual active search in which we decompose the search policy into a prediction and search modules. This decomposition enables us to combine supervised and reinforcement learning for training, and make use of supervised learning even during execution. Moreover, we propose a novel meta-learning framework to jointly learn a policy and initialization parameters for the supervised prediction module. Our findings demonstrate the significance of the proposed frameworks for conducting efficient search, particularly in real-world situations where search tasks may differ significantly from those utilized during policy training. We hope our framework will find its applicability in many practical scenarios ranging from human trafficking to animal poaching.

## Acknowledgments

This research was partially supported by the NSF (IIS-1905558, IIS-1903207, and IIS-2214141), ARO (W911NF-18-1-0208), Amazon, NVIDIA, and the Taylor Geospatial Institute managed by Saint Louis University.

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

# A Partially Supervised Reinforcement Learning Framework for Visual Active Search: Supplementary Material

## A  Policy Network Architecture and Hyperparameter Details

Recall that the policy network $\pi$ is composed of two parts : (1) a task specific prediction module, and (2) a task-agnostic search module. The task specific prediction module consists of an encoder $e(x; \eta)$ that maps the aerial image $x$ to a low-dimensional latent feature representation $z$, and a grid prediction network $p(z, o; \kappa)$ that predicts the probabilities of grids containing a target by leveraging the latent semantic feature $z$ and the outcomes of previous search queries $o$. Note that the task specific prediction module is represented as $f(x, o, \theta) = p(z = e(x; \eta), o; \kappa)$, where $\theta = (\eta, \kappa)$. Following [6], we use frozen ResNet-34, pre-trained on ImageNet, followed by a learnable $1 \times 1$ convolution layer with a ReLU activation as a feature extraction component of the task specific prediction module that we refer as encoder $e(.)$. We then combine the latent semantic feature $z$ with the previous query information $o$. We apply the tiling operation in order to convert $o$ into a representation with the same dimensions as the extracted features $z$, enabling us to effectively apply channel-wise concatenation of latent image feature and auxiliary state feature while preserving the grid specific spatial and query related information. This combined representation is then fed to a grid prediction network comprises of a $1 \times 1$ convolution layer, flattening, and a MLP block consists of 2 fully connected layer with ReLU activations. Note that the output of grid prediction network is of dimension $N$. We finally apply sigmoid activation to each output neuron to convert them into a probability value representing the probability of the grids containing target. The proposed policy architecture is depicted in figure 2 of the main paper.

We re-shape the output of task specific prediction module by converting it back from 1D to 2D of shape $(m \times n) = N$ before feeding it to the task agnostic search module $g(.)$ that takes the following three inputs: (1) the reshaped 2D output of the task specific prediction module, which is the probabilities of grids containing target; (2) the remaining search budget $B$, which is a scalar but we apply tiling to the scalar budget $B$ to transform it to match the size of the reshaped 2D output of the task specific prediction module; (3) we also apply the tiling operation to $o$ in a way that allows us to concatenate the features $(z, o, B)$ along the channels dimension to finally obtain the combined representation that serves as a input to task agnostic search module. The task agnostic search module is composed of a flattening, a MLP block consists of 2 fully connected layer with ReLU activations, and a final softmax layer to convert the output to a probability distribution that guides us in selecting the grid to query next.

In Table 8, we detail the architecture of task specific prediction module ($f$) of PSVAS policy network. In Table 9, we detail the architecture of task agnostic search module ($g$) of PSVAS policy network. Note that, the task specific prediction module and task agnostic search module remains unchanged in MPS-VAS framework.

Table 8: Task Specific Prediction Module Architecture with number of grid cell $N = (m \times n)$

| Layers | Configuration | o/p Feature Map size |
|---|---|---|
| Input | RGB Image | $3 \times 3500 \times 3500$ |
| Encoder | ResNet-34 | $512 \times 14 \times 14$ |
| Conv1 | Channel:N; kernel size:$1 \times 1$ | $N \times 14 \times 14$ |
| 2D MaxPool | Pooling size:$2 \times 2$ | $N \times 7 \times 7$ |
| Tile1 | Grid State ($o$) | $N \times 7 \times 7$ |
| Channelwise Concat | Conv1,Tile1 | $(2N) \times 7 \times 7$ |
| Conv2 | Channel:3; kernel size: $1 \times 1$ | $3 \times 7 \times 7$ |
| Flattened | Conv2 | 147 |
| FC1+ReLU | $(147-> 2N)$ | 2N |
| FC2+Sigmoid | $(2N-> N)$ | N |

Table 9: Task Agnostic Search Module Architecture with number of grid cell $N = (m \times n)$

| Layers | Configuration | o/p Feature Map size |
|---|---|---|
| Input 1 | 2D Reshape of Task Specific Prediction Module Output | $1 \times m \times n$ |
| Input 2: Tile2 | Grid State ($o$) | $1 \times m \times n$ |
| Input 3: Tile3 | Query Budget Left ($B$) | $1 \times m \times n$ |
| Input: Channelwise Concat | Input 1, Input 2, Input 3 | $(3) \times m \times n$ |
| Flattened | Input: Channelwise Concat | $K = (3) \times m \times n$ |
| FC1+ReLU | $(K- > 2N)$ | 2N |
| FC2+Softmax | $(2N- > N)$ | N |

In MPS-VAS-MQ framework, the network architecture of task specific prediction module remains unaltered, but the additional dependence of task agnostic search module ($g$) on $\psi$ enforce a slight modification of its architecture as detailed in Table 10.

Table 10: Task Agnostic Search Module Architecture in multi query setting with number of grid cell $N = (m \times n)$

| Layers | Configuration | o/p Feature Map size |
|---|---|---|
| Input 1 | 2D Reshape of Task Specific Prediction Module Output | $1 \times m \times n$ |
| Input 2: Tile2 | Grid State ($o$) | $1 \times m \times n$ |
| Input 3: Tile3 | Query Budget Left ($B$) | $1 \times m \times n$ |
| Input 4: Tile4 | Encoded Locations of the queried Grid cells ($\psi$) | $1 \times m \times n$ |
| Input: Channelwise Concat | Input 1, Input 2, Input 3, Input 4 | $(4) \times m \times n$ |
| Flattened | Input: Channelwise Concat | $D = (4) \times m \times n$ |
| FC1+ReLU | $(D- > 2N)$ | 2N |
| FC2+Softmax | $(2N- > N)$ | N |

We use a learning rate of $10^{-4}$, batch size of 16, number of training epochs 200, and the Adam optimizer to train the policy network in all experimental settings. During Inference, in all experimental settings, we update the parameters of task specific prediction module $f$ after each query step using a learning rate of $10^{-4}$ and the Adam optimizer. We use 1 NVidia A100 and 3 GeForce GTX 1080Ti GPU servers for all our experiments.

## B    Results with Uniform Query Cost

### B.1    Single Query Setting

Here we present the results by considering a setting with a single query resource and query costs $c(i, j) = 1$ for all $i, j$, where $\mathcal{C}$ is the number of queries. We evaluate PSVAS and MPS-VAS on the xView dataset with varying search budget $\mathcal{C} \in \{12, 15, 18\}$ and the number of grid cells $N = 49$. We train the policy with *small car* as the target and test the performance of the policy with the following target classes : *Small Car (SC)*, *Helicopter*, *Sail Boat (SB)*, *Construction Cite (CC)*, *Building*, and *Helipad*. The results are presented in Table 11. We observe noticeable improvement in performance of the proposed PSVAS approach compared to all baselines in each different target setting, ranging from approximately 0.50 to 52.0% relative to the most competitive E2EVAS baseline. In Table 12, we report the results on DOTA dataset with $N = 64$. In this setting, we train the policy with *large vehicle* as the target and evaluate the performance with the following target classes : *Ship*, *large vehicle* (LV), *Harbor*, *Helicopter*, *Plane*, and *Roundabout*. Here, we notice significant improvement in performance of PSVAS compared to all the baselines including E2EVAS, ranging from approximately 3.5 to 25.0%. The effectiveness of the PSVAS framework becomes evident as it allows us to efficiently update the task-specific prediction module $f$ by leveraging the crucial supervised information. We

also observe a consistent trend, i.e., the performance of MPS-VAS is significantly better than PSVAS across different target settings, ranging from approximately $0.6$ to $60.0\%$. The significance of the MPS-VAS framework becomes apparent when deploying visual active search in scenarios where the search tasks differ substantially from those encountered during training.

Table 11: **ANT** comparisons when trained with *small car* as target on xView in single-query setting.

| | Test with Helicopter as Target | | | Test with SB as Target | | | Test with Building as Target | | |
|---|---|---|---|---|---|---|---|---|---|
| Method | $\mathcal{C} = 12$ | $\mathcal{C} = 15$ | $\mathcal{C} = 18$ | $\mathcal{C} = 12$ | $\mathcal{C} = 15$ | $\mathcal{C} = 18$ | $\mathcal{C} = 12$ | $\mathcal{C} = 15$ | $\mathcal{C} = 18$ |
| RS | 0.41 | 0.52 | 0.65 | 0.62 | 0.83 | 0.93 | 4.74 | 6.05 | 7.11 |
| GC | 0.44 | 0.59 | 0.78 | 0.73 | 0.92 | 0.99 | 5.45 | 6.53 | 7.65 |
| GS [15] | 0.47 | 0.61 | 0.84 | 0.78 | 0.96 | 1.03 | 5.68 | 6.87 | 8.01 |
| AL [13] | 0.43 | 0.59 | 0.77 | 0.72 | 0.90 | 0.97 | 5.44 | 6.53 | 7.63 |
| AS [9] | 0.44 | 0.57 | 0.75 | 0.70 | 0.89 | 0.96 | 5.32 | 6.38 | 7.44 |
| E2EVAS [6] | 0.50 | 0.63 | 0.92 | 0.83 | 1.06 | 1.10 | 7.29 | 8.78 | 10.14 |
| OnlineTTA[6] | 0.50 | 0.64 | 0.93 | 0.84 | 1.06 | 1.11 | 7.29 | 8.79 | 10.15 |
| **PSVAS** | **0.91** | **0.95** | **1.08** | **0.97** | **1.13** | **1.37** | **7.30** | **8.81** | **10.28** |
| **MPS-VAS** | **1.04** | **1.13** | **1.21** | **1.23** | **1.50** | **1.74** | **7.32** | **8.83** | **10.33** |
| | Test with CC as Target | | | Test with SC as Target | | | Test with Helipad as Target | | |
| Method | $\mathcal{C} = 12$ | $\mathcal{C} = 15$ | $\mathcal{C} = 18$ | $\mathcal{C} = 12$ | $\mathcal{C} = 15$ | $\mathcal{C} = 18$ | $\mathcal{C} = 12$ | $\mathcal{C} = 15$ | $\mathcal{C} = 18$ |
| RS | 1.19 | 1.54 | 1.81 | 3.62 | 4.57 | 5.51 | 0.38 | 0.47 | 0.61 |
| GC | 1.42 | 1.86 | 2.19 | 4.06 | 4.98 | 6.03 | 0.51 | 0.65 | 0.83 |
| GS [15] | 1.61 | 2.01 | 2.33 | 4.59 | 5.54 | 6.71 | 0.56 | 0.74 | 0.96 |
| AL [13] | 1.41 | 1.85 | 2.17 | 4.03 | 4.96 | 6.02 | 0.51 | 0.63 | 0.82 |
| AS [9] | 1.40 | 1.74 | 2.09 | 3.96 | 4.92 | 5.97 | 0.47 | 0.59 | 0.77 |
| E2EVAS [6] | 1.74 | 2.10 | 2.46 | 5.80 | 7.02 | 8.15 | 0.90 | 1.06 | 1.23 |
| OnlineTTA[6] | 1.75 | 2.12 | 2.46 | 5.81 | 7.03 | 8.15 | 0.91 | 1.06 | 1.23 |
| **PSVAS** | **1.86** | **2.25** | **2.61** | **5.94** | **7.10** | **8.19** | **1.02** | **1.09** | **1.26** |
| **MPS-VAS** | **1.97** | **2.35** | **2.76** | **5.99** | **7.16** | **8.24** | **1.07** | **1.16** | **1.37** |

Table 12: **ANT** comparisons when trained with *large vehicle* as target on DOTA in single-query setting.

| | Test with Ship as Target | | | Test with LV as Target | | | Test with Harbor as Target | | |
|---|---|---|---|---|---|---|---|---|---|
| Method | $\mathcal{C} = 12$ | $\mathcal{C} = 15$ | $\mathcal{C} = 18$ | $\mathcal{C} = 12$ | $\mathcal{C} = 15$ | $\mathcal{C} = 18$ | $\mathcal{C} = 12$ | $\mathcal{C} = 15$ | $\mathcal{C} = 18$ |
| Random | 2.41 | 3.02 | 3.95 | 3.40 | 4.03 | 5.14 | 3.17 | 3.93 | 4.78 |
| GC | 2.82 | 3.44 | 4.27 | 3.87 | 4.59 | 5.55 | 3.48 | 4.25 | 4.98 |
| GS[15] | 2.96 | 3.59 | 4.48 | 3.99 | 4.77 | 5.67 | 3.62 | 4.40 | 5.07 |
| AL[13] | 2.81 | 3.42 | 4.26 | 3.85 | 4.54 | 5.51 | 3.47 | 4.25 | 4.97 |
| AS[9] | 2.57 | 3.27 | 4.03 | 3.61 | 4.12 | 5.26 | 3.35 | 4.16 | 4.92 |
| E2EVAS[6] | 3.57 | 4.42 | 5.15 | 6.30 | 7.65 | 8.90 | 4.28 | 5.21 | 6.09 |
| OnlineTTA[6] | 3.57 | 4.43 | 5.15 | 6.31 | 7.67 | 8.90 | 4.30 | 5.22 | 6.10 |
| **PSVAS** | **3.60** | **4.51** | **5.23** | **6.50** | **7.86** | **9.22** | **4.61** | **5.72** | **6.87** |
| **MPS-VAS** | **3.79** | **4.75** | **5.58** | **6.51** | **7.88** | **9.24** | **4.90** | **6.23** | **7.38** |
| | Test with Helicopter as Target | | | Test with Plane as Target | | | Test with Roundabout as Target | | |
| Method | $\mathcal{C} = 12$ | $\mathcal{C} = 15$ | $\mathcal{C} = 18$ | $\mathcal{C} = 12$ | $\mathcal{C} = 15$ | $\mathcal{C} = 18$ | $\mathcal{C} = 12$ | $\mathcal{C} = 15$ | $\mathcal{C} = 18$ |
| Random | 0.66 | 0.73 | 0.82 | 2.91 | 3.94 | 4.74 | 2.66 | 3.59 | 4.37 |
| GC | 0.71 | 0.82 | 0.89 | 3.22 | 4.35 | 5.07 | 2.93 | 3.81 | 4.59 |
| GS[15] | 0.75 | 0.87 | 0.97 | 3.47 | 4.56 | 5.25 | 2.99 | 3.96 | 4.73 |
| AL[13] | 0.70 | 0.81 | 0.88 | 3.22 | 4.34 | 5.07 | 2.93 | 3.79 | 4.59 |
| AS[9] | 0.68 | 0.78 | 0.86 | 3.16 | 4.21 | 4.97 | 2.82 | 3.74 | 4.51 |
| E2EVAS[6] | 0.78 | 0.96 | 1.18 | 4.02 | 5.07 | 5.90 | 4.00 | 5.05 | 5.88 |
| OnlineTTA[6] | 0.78 | 0.97 | 1.19 | 4.02 | 5.07 | 5.91 | 4.01 | 5.06 | 5.88 |
| **PSVAS** | **0.95** | **1.21** | **1.49** | **4.33** | **5.32** | **6.44** | **4.33** | **5.36** | **6.41** |
| **MPS-VAS** | **1.10** | **1.37** | **1.67** | **4.52** | **5.58** | **6.75** | **4.51** | **5.56** | **6.73** |

## B.2  Multi Query Setting

In Table 13, we present the results of MPS-VAS-MQ and compare its performance with MPS-VAS-TOPK with varying search budget $\mathcal{C} \in \{12, 15, 18\}$ and the number of grid cell N=49. Here, we train the policy with *small car* as the target and evaluate the performance of the policy with the following target classes : *Small Car* (SC), *Helicopter*, *Sail Boat* (SB), *Construction Cite* (CC), *Building*, and *Helipad*. In table 14, we present similar results with the number of grid cell $N = 64$. In this setting, we train the policy with *Large Vehicle* as the target and evaluate the policy with the

following target classes: *Ship*, *Large Vehicle* (LV), *Harbor*, *Helicopter*, *Plane*, and *Roundabout* (RB). We consider $K = 3$ in all these experiments. We observe a consistent improvement in performance of MPS-VAS-MQ over MPS-VAS-TOPK across different target setting, ranging from approximately 0.1 to 15%. The experimental results indicate that there are additional benefits in learning to capture the interdependence in greedy search decisions.

Table 13: **ANT** comparisons when trained with *small car* as target on xView in multi-query setting.

| | Test with Helicopter as Target | | | Test with SB as Target | | | Test with Building as Target | | |
|---|---|---|---|---|---|---|---|---|---|
| Method | $\mathcal{C} = 12$ | $\mathcal{C} = 15$ | $\mathcal{C} = 18$ | $\mathcal{C} = 12$ | $\mathcal{C} = 15$ | $\mathcal{C} = 18$ | $\mathcal{C} = 12$ | $\mathcal{C} = 15$ | $\mathcal{C} = 18$ |
| MPS-VAS-TOPK | 0.71 | 0.85 | 1.04 | 1.10 | 1.23 | 1.47 | 7.07 | 8.60 | 9.98 |
| **MPS-VAS-MQ** | **0.75** | **0.88** | **1.08** | **1.14** | **1.41** | **1.53** | **7.31** | **8.81** | **10.21** |
| | Test with CC as Target | | | Test with SC as Target | | | Test with Helipad as Target | | |
| Method | $\mathcal{C} = 12$ | $\mathcal{C} = 15$ | $\mathcal{C} = 18$ | $\mathcal{C} = 12$ | $\mathcal{C} = 15$ | $\mathcal{C} = 18$ | $\mathcal{C} = 12$ | $\mathcal{C} = 15$ | $\mathcal{C} = 18$ |
| MPS-VAS-TOPK | 1.89 | 2.09 | 2.50 | 5.78 | 6.92 | 7.98 | 0.82 | 0.93 | 1.10 |
| **MPS-VAS-MQ** | **1.95** | **2.27** | **2.68** | **5.97** | **7.09** | **8.16** | **1.03** | **1.09** | **1.23** |

Table 14: **ANT** comparisons when trained with *large vehicle* as target on DOTA in multi-query setting.

| | Test with Ship as Target | | | Test with LV as Target | | | Test with Harbor as Target | | |
|---|---|---|---|---|---|---|---|---|---|
| Method | $\mathcal{C} = 12$ | $\mathcal{C} = 15$ | $\mathcal{C} = 18$ | $\mathcal{C} = 12$ | $\mathcal{C} = 15$ | $\mathcal{C}18$ | $\mathcal{C} = 12$ | $\mathcal{C} = 15$ | $\mathcal{C} = 18$ |
| MPS-VAS-TOPK | 3.72 | 4.66 | 5.49 | 6.09 | 7.29 | 8.54 | 4.76 | 6.14 | 7.31 |
| **MPS-VAS-MQ** | **3.74** | **4.69** | **5.54** | **6.36** | **7.64** | **8.79** | **4.78** | **6.20** | **7.32** |
| | Test with Helicopter as Target | | | Test with Plane as Target | | | Test with RB as Target | | |
| Method | $\mathcal{C} = 12$ | $\mathcal{C} = 15$ | $\mathcal{C} = 18$ | $\mathcal{C} = 12$ | $\mathcal{C} = 15$ | $\mathcal{C} = 18$ | $\mathcal{C} = 12$ | $\mathcal{C} = 15$ | $\mathcal{C} = 18$ |
| MPS-VAS-TOPK | 0.88 | 1.05 | 1.24 | 3.95 | 5.48 | 6.69 | 4.32 | 5.45 | 6.45 |
| **MPS-VAS-MQ** | **0.90** | **1.06** | **1.30** | **4.02** | **5.49** | **6.73** | **4.39** | **5.47** | **6.49** |

## C   Results with Different Number of grid cells

Here, we present the results of PSVAS and MPS-VAS and compare the performance with the most competitive E2EVAS approach for different choices of $N$.

### C.1   Results with Number of Grid cell $N = 99$

In this setting, we train the policy with *small car* as the target and evaluate the performance of the policy with the following target classes : *Small Car* (SC), *Helicopter*, *Sail Boat* (SB), *Construction Cite* (CC), *Building*, and *Helipad*. In Table 15, we present the results with *Manhattan distance based query cost* in single query setting. The similar results with multi query setting are presented in Table 16. In Table 17 and 18, we present the results with *uniform query cost* in single and multi query setting respectively. We notice a very similar trend in performance as observed in the settings with other choices of $N$. Specifically, We observe PSVAS significantly outperforms E2EVAS across different target settings, and MPS-VAS further improves the search performance universally. These results highlights the effectiveness of our proposed PSVAS and MPS-VAS framework for visual active search in practical scenarios when search tasks differ from those that are used for policy training.

### C.2   Results with Number of Grid cell $N = 36$

In this setting, we train the policy with *large vehicle* as the target and evaluate the performance with the following target classes : *Ship*, *large vehicle* (LV), *Harbor*, *Helicopter*, *Plane*, and *Roundabout*. In Table 19, we present the results with Manhattan distance based query cost in single query setting. The results with multi query setting are presented in Table 20. In Table 21 and 22, we present the the results with uniform query cost in single and multi query setting respectively. We observe a consistent performance trend across various target settings. Specifically, PSVAS consistently outperforms E2EVAS in different target settings, and the introduction of MPS-VAS further enhances the search performance across the board. These results emphasize the effectiveness of our proposed PSVAS and

Table 15: **ANT** comparisons when trained with *small car* as target on xView in single-query setting.

| | Test with Helicopter as Target | | | Test with SB as Target | | | Test with Building as Target | | |
|---|---|---|---|---|---|---|---|---|---|
| Method | $\mathcal{C}=25$ | $\mathcal{C}=50$ | $\mathcal{C}=75$ | $\mathcal{C}=25$ | $\mathcal{C}=50$ | $\mathcal{C}=75$ | $\mathcal{C}=25$ | $\mathcal{C}=50$ | $\mathcal{C}=75$ |
| RS | 0.01 | 0.09 | 0.14 | 0.23 | 0.34 | 0.61 | 1.41 | 2.51 | 3.84 |
| E2EVAS [6] | 0.17 | 0.30 | 0.39 | 0.65 | 1.03 | 1.34 | 3.32 | 5.37 | 7.05 |
| OnlineTTA[6] | 0.17 | 0.31 | 0.40 | 0.66 | 1.03 | 1.34 | 3.32 | 5.39 | 7.07 |
| **PSVAS** | **0.39** | **0.48** | **0.65** | **0.71** | **1.07** | **1.35** | **4.31** | **6.97** | **9.12** |
| **MPS-VAS** | **0.45** | **0.55** | **0.69** | **0.75** | **1.08** | **1.37** | **4.42** | **7.18** | **9.35** |

| | Test with CC as Target | | | Test with SC as Target | | | Test with Helipad as Target | | |
|---|---|---|---|---|---|---|---|---|---|
| Method | $\mathcal{C}=25$ | $\mathcal{C}=50$ | $\mathcal{C}=75$ | $\mathcal{C}=25$ | $\mathcal{C}=50$ | $\mathcal{C}=75$ | $\mathcal{C}=25$ | $\mathcal{C}=50$ | $\mathcal{C}=75$ |
| RS | 0.32 | 0.56 | 0.87 | 1.10 | 2.15 | 2.96 | 0.12 | 0.19 | 0.29 |
| E2EVAS [6] | 0.61 | 1.03 | 1.41 | 2.72 | 4.42 | 5.78 | 0.39 | 0.44 | 0.56 |
| OnlineTTA[6] | 0.63 | 1.04 | 1.41 | 2.72 | 4.43 | 5.79 | 0.39 | 0.45 | 0.56 |
| **PSVAS** | **0.98** | **1.72** | **2.19** | **3.12** | **5.01** | **6.40** | **0.46** | **0.59** | **0.74** |
| **MPS-VAS** | **1.01** | **1.77** | **2.28** | **3.34** | **5.31** | **6.74** | **0.51** | **0.66** | **0.86** |

Table 16: **ANT** comparisons when trained with *small car* as target on xView in multi-query setting.

| | Test with Helicopter as Target | | | Test with SB as Target | | | Test with Building as Target | | |
|---|---|---|---|---|---|---|---|---|---|
| Method | $\mathcal{C}=25$ | $\mathcal{C}=50$ | $\mathcal{C}=75$ | $\mathcal{C}=25$ | $\mathcal{C}=50$ | $\mathcal{C}=75$ | $\mathcal{C}=25$ | $\mathcal{C}=50$ | $\mathcal{C}=75$ |
| MPS-VAS-TOPK | 0.40 | 0.51 | 0.62 | 0.69 | 0.98 | 1.30 | 4.29 | 6.84 | 8.66 |
| **MPS-VAS-MQ** | **0.42** | **0.53** | **0.66** | **0.71** | **1.03** | **1.32** | **4.33** | **6.95** | **8.78** |

| | Test with CC as Target | | | Test with SC as Target | | | Test with Helipad as Target | | |
|---|---|---|---|---|---|---|---|---|---|
| Method | $\mathcal{C}=25$ | $\mathcal{C}=50$ | $\mathcal{C}=75$ | $\mathcal{C}=25$ | $\mathcal{C}=50$ | $\mathcal{C}=75$ | $\mathcal{C}=25$ | $\mathcal{C}=50$ | $\mathcal{C}=75$ |
| MPS-VAS-TOPK | 0.96 | 1.53 | 2.12 | 3.19 | 5.09 | 6.47 | 0.45 | 0.59 | 0.77 |
| **MPS-VAS-MQ** | **0.98** | **1.65** | **2.17** | **3.25** | **5.12** | **6.55** | **0.47** | **0.61** | **0.82** |

Table 17: **ANT** comparisons when trained with *small car* as target on xView in single-query setting.

| | Test with Helicopter as Target | | | Test with SB as Target | | | Test with Building as Target | | |
|---|---|---|---|---|---|---|---|---|---|
| Method | $\mathcal{C}=12$ | $\mathcal{C}=15$ | $\mathcal{C}=18$ | $\mathcal{C}=12$ | $\mathcal{C}=15$ | $\mathcal{C}=18$ | $\mathcal{C}=12$ | $\mathcal{C}=15$ | $\mathcal{C}=18$ |
| RS | 0.22 | 0.31 | 0.38 | 0.48 | 0.55 | 0.63 | 3.43 | 4.25 | 4.97 |
| E2EVAS [6] | 0.31 | 0.39 | 0.43 | 0.80 | 1.05 | 1.30 | 5.23 | 6.37 | 7.41 |
| OnlineTTA[6] | 0.31 | 0.40 | 0.44 | 0.80 | 1.06 | 1.31 | 5.24 | 6.38 | 7.43 |
| **PSVAS** | **0.43** | **0.48** | **0.51** | **0.83** | **1.09** | **1.33** | **5.34** | **6.41** | **7.52** |
| **MPS-VAS** | **0.47** | **0.50** | **0.54** | **0.84** | **1.11** | **1.39** | **5.44** | **6.69** | **7.75** |

| | Test with CC as Target | | | Test with SC as Target | | | Test with Helipad as Target | | |
|---|---|---|---|---|---|---|---|---|---|
| Method | $\mathcal{C}=12$ | $\mathcal{C}=15$ | $\mathcal{C}=18$ | $\mathcal{C}=12$ | $\mathcal{C}=15$ | $\mathcal{C}=18$ | $\mathcal{C}=12$ | $\mathcal{C}=15$ | $\mathcal{C}=18$ |
| RS | 0.78 | 1.02 | 1.17 | 3.12 | 3.61 | 4.45 | 0.25 | 0.33 | 0.41 |
| E2EVAS [6] | 0.98 | 1.29 | 1.47 | 4.61 | 5.64 | 6.55 | 0.44 | 0.46 | 0.56 |
| OnlineTTA[6] | 0.99 | 1.32 | 1.50 | 4.62 | 5.64 | 6.56 | 0.45 | 0.47 | 0.56 |
| **PSVAS** | **1.28** | **1.64** | **1.86** | **4.74** | **5.72** | **6.75** | **0.53** | **0.59** | **0.78** |
| **MPS-VAS** | **1.39** | **1.69** | **2.05** | **4.81** | **5.93** | **6.96** | **0.61** | **0.66** | **0.83** |

Table 18: **ANT** comparisons when trained with *small car* as target on xView in multi-query setting.

| | Test with Helicopter as Target | | | Test with SB as Target | | | Test with Building as Target | | |
|---|---|---|---|---|---|---|---|---|---|
| Method | $\mathcal{C}=12$ | $\mathcal{C}=15$ | $\mathcal{C}=18$ | $\mathcal{C}=12$ | $\mathcal{C}=15$ | $\mathcal{C}=18$ | $\mathcal{C}=12$ | $\mathcal{C}=15$ | $\mathcal{C}=18$ |
| MPS-VAS-TOPK | 0.41 | 0.43 | 0.48 | 0.78 | 1.01 | 1.26 | 4.91 | 6.07 | 7.02 |
| **MPS-VAS-MQ** | **0.42** | **0.46** | **0.51** | **0.81** | **1.05** | **1.32** | **5.02** | **6.21** | **7.18** |

| | Test with CC as Target | | | Test with SC as Target | | | Test with Helipad as Target | | |
|---|---|---|---|---|---|---|---|---|---|
| Method | $\mathcal{C}=12$ | $\mathcal{C}=15$ | $\mathcal{C}=18$ | $\mathcal{C}=12$ | $\mathcal{C}=15$ | $\mathcal{C}=18$ | $\mathcal{C}=12$ | $\mathcal{C}=15$ | $\mathcal{C}=18$ |
| MPS-VAS-TOPK | 1.22 | 1.41 | 1.82 | 4.29 | 5.63 | 6.59 | 0.54 | 0.59 | 0.78 |
| **MPS-VAS-MQ** | **1.26** | **1.53** | **1.98** | **4.38** | **5.74** | **6.68** | **0.57** | **0.61** | **0.79** |

MPS-VAS framework for visual active search in real-world scenarios where the search tasks differ from the ones used during policy training.

Table 19: **ANT** comparisons when trained with *LV* as target on DOTA in single-query setting.

| | Test with Ship as Target | | | Test with LV as Target | | | Test with Harbor as Target | | |
|---|---|---|---|---|---|---|---|---|---|
| Method | $\mathcal{C}=25$ | $\mathcal{C}=50$ | $\mathcal{C}=75$ | $\mathcal{C}=25$ | $\mathcal{C}=50$ | $\mathcal{C}=75$ | $\mathcal{C}=25$ | $\mathcal{C}=50$ | $\mathcal{C}=75$ |
| RS | 1.45 | 3.17 | 4.30 | 1.79 | 3.50 | 5.10 | 2.35 | 4.34 | 6.76 |
| E2EVAS [6] | 2.69 | 4.50 | 5.88 | 4.63 | 6.79 | 8.07 | 4.22 | 6.92 | 9.06 |
| OnlineTTA[6] | 2.70 | 4.52 | 5.89 | 4.63 | 6.80 | 8.07 | 4.22 | 6.93 | 9.08 |
| **PSVAS** | **3.19** | **4.83** | **6.34** | **4.69** | **6.94** | **8.12** | **4.95** | **7.56** | **9.51** |
| **MPS-VAS** | **3.42** | **5.19** | **6.73** | **4.80** | **7.08** | **8.23** | **5.02** | **8.04** | **9.91** |
| | Test with Helicopter as Target | | | Test with Plane as Target | | | Test with RB as Target | | |
| Method | $\mathcal{C}=25$ | $\mathcal{C}=50$ | $\mathcal{C}=75$ | $\mathcal{C}=25$ | $\mathcal{C}=50$ | $\mathcal{C}=75$ | $\mathcal{C}=25$ | $\mathcal{C}=50$ | $\mathcal{C}=75$ |
| RS | 0.60 | 1.27 | 1.96 | 2.33 | 4.34 | 6.62 | 0.64 | 1.06 | 1.80 |
| E2EVAS [6] | 1.00 | 2.07 | 2.66 | 4.57 | 7.23 | 9.14 | 1.56 | 2.28 | 2.72 |
| OnlineTTA[6] | 1.00 | 2.07 | 2.68 | 4.57 | 7.25 | 9.16 | 1.56 | 2.28 | 2.73 |
| **PSVAS** | **1.53** | **2.33** | **2.84** | **5.09** | **7.64** | **9.41** | **1.87** | **2.34** | **2.76** |
| **MPS-VAS** | **1.80** | **2.60** | **3.03** | **5.17** | **7.83** | **10.02** | **1.96** | **2.76** | **3.19** |

Table 20: **ANT** comparisons when trained with *LV* as target on DOTA in multi-query setting.

| | Test with Ship as Target | | | Test with LV as Target | | | Test with Harbor as Target | | |
|---|---|---|---|---|---|---|---|---|---|
| Method | $\mathcal{C}=25$ | $\mathcal{C}=50$ | $\mathcal{C}=75$ | $\mathcal{C}=25$ | $\mathcal{C}=50$ | $\mathcal{C}=75$ | $\mathcal{C}=25$ | $\mathcal{C}=50$ | $\mathcal{C}=75$ |
| MPS-VAS-TOPK | 3.33 | 5.14 | 6.70 | 4.64 | 6.83 | 7.79 | 4.96 | 7.91 | 9.75 |
| **MPS-VAS-MQ** | **3.38** | **5.17** | **6.71** | **4.65** | **6.92** | **8.00** | **4.99** | **7.98** | **9.83** |
| | Test with Helicopter as Target | | | Test with Plane as Target | | | Test with RB as Target | | |
| Method | $\mathcal{C}=25$ | $\mathcal{C}=50$ | $\mathcal{C}=75$ | $\mathcal{C}=25$ | $\mathcal{C}=50$ | $\mathcal{C}=75$ | $\mathcal{C}=25$ | $\mathcal{C}=50$ | $\mathcal{C}=75$ |
| MPS-VAS-TOPK | 1.34 | 2.42 | 2.88 | 5.08 | 7.63 | 9.66 | 1.76 | 2.68 | 3.02 |
| **MPS-VAS-MQ** | **1.37** | **2.43** | **2.91** | **5.15** | **7.75** | **9.95** | **1.82** | **2.72** | **3.11** |

Table 21: **ANT** comparisons when trained with *LV* as target on DOTA in single-query setting.

| | Test with Ship as Target | | | Test with LV as Target | | | Test with Harbor as Target | | |
|---|---|---|---|---|---|---|---|---|---|
| Method | $\mathcal{C}=12$ | $\mathcal{C}=15$ | $\mathcal{C}=18$ | $\mathcal{C}=12$ | $\mathcal{C}=15$ | $\mathcal{C}=18$ | $\mathcal{C}=12$ | $\mathcal{C}=15$ | $\mathcal{C}=18$ |
| RS | 2.92 | 3.34 | 3.99 | 3.44 | 4.08 | 5.19 | 4.17 | 5.04 | 5.92 |
| E2EVAS [6] | 3.34 | 4.15 | 4.77 | 5.14 | 6.05 | 7.00 | 5.38 | 6.51 | 7.54 |
| OnlineTTA[6] | 3.36 | 4.15 | 4.79 | 5.14 | 6.06 | 7.01 | 5.40 | 6.52 | 7.55 |
| **PSVAS** | **3.48** | **4.37** | **5.15** | **5.23** | **6.08** | **7.12** | **5.57** | **6.69** | **7.78** |
| **MPS-VAS** | **3.85** | **4.69** | **5.38** | **5.25** | **6.11** | **7.14** | **5.71** | **6.95** | **8.15** |
| | Test with Helicopter as Target | | | Test with Plane as Target | | | Test with RB as Target | | |
| Method | $\mathcal{C}=12$ | $\mathcal{C}=15$ | $\mathcal{C}=18$ | $\mathcal{C}=12$ | $\mathcal{C}=15$ | $\mathcal{C}=18$ | $\mathcal{C}=12$ | $\mathcal{C}=15$ | $\mathcal{C}=18$ |
| RS | 1.03 | 1.52 | 1.77 | 4.05 | 5.11 | 6.12 | 1.25 | 1.54 | 1.91 |
| E2EVAS [6] | 1.50 | 1.87 | 2.13 | 5.47 | 6.59 | 7.65 | 1.87 | 2.17 | 2.47 |
| OnlineTTA[6] | 1.50 | 1.88 | 2.16 | 5.47 | 6.61 | 7.68 | todo | todo | todo |
| **PSVAS** | **1.77** | **2.23** | **2.50** | **5.54** | **6.65** | **7.66** | **2.03** | **2.32** | **2.65** |
| **MPS-VAS** | **2.10** | **2.57** | **2.77** | **5.73** | **6.87** | **7.90** | **2.12** | **2.66** | **2.99** |

Table 22: **ANT** comparisons when trained with *LV* as target on DOTA in multi-query setting.

| | Test with Ship as Target | | | Test with LV as Target | | | Test with Harbor as Target | | |
|---|---|---|---|---|---|---|---|---|---|
| Method | $\mathcal{C}=12$ | $\mathcal{C}=15$ | $\mathcal{C}=18$ | $\mathcal{C}=12$ | $\mathcal{C}=15$ | $\mathcal{C}=18$ | $\mathcal{C}=12$ | $\mathcal{C}=15$ | $\mathcal{C}=18$ |
| MPS-VAS-TOPK | 3.84 | 4.64 | 5.28 | 5.14 | 6.01 | 6.51 | 5.65 | 6.84 | 7.93 |
| **MPS-VAS-MQ** | **3.81** | **4.64** | **5.35** | **5.22** | **6.05** | **6.68** | **5.66** | **6.89** | **8.04** |
| | Test with Helicopter as Target | | | Test with Plane as Target | | | Test with RB as Target | | |
| Method | $\mathcal{C}=12$ | $\mathcal{C}=15$ | $\mathcal{C}=18$ | $\mathcal{C}=12$ | $\mathcal{C}=15$ | $\mathcal{C}=18$ | $\mathcal{C}=12$ | $\mathcal{C}=15$ | $\mathcal{C}=18$ |
| MPS-VAS-TOPK | 1.39 | 1.91 | 2.27 | 5.64 | 6.79 | 7.71 | 2.01 | 2.43 | 2.68 |
| **MPS-VAS-MQ** | **1.43** | **1.96** | **2.33** | **5.65** | **6.83** | **7.80** | **2.08** | **2.49** | **2.81** |

# D  Effect of Inference Time Adaptation of Task Specific Prediction Module on Search Performance

## D.1  Effect on PSVAS Framework

First, we analyze the impact of inference time adaptation of task specific prediction module on PSVAS framework across different target settings. To this end, we first train a policy using our proposed PSVAS approach and then during inference we freeze the task specific prediction module along with task agnostic search module unlike PSVAS approach. We call the resulting policy as PSVAS-F. In Table 23, we compare the search performance of PSVAS and PSVAS-F with number of grid cell $N = 36$ across different target settings. In Table 24, we present similar results with number of grid cell $N = 49$. We observe a significant improvement in performance of PSVAS compared to PSVAS-F across different target settings, justifying the importance of inference time adaptation of task specific prediction module after every query.

Table 23: **ANT** comparisons when trained with *LV* as target on DOTA in single-query setting.

| | Test with Ship as Target | | | Test with LV as Target | | | Test with Harbor as Target | | |
| --- | --- | --- | --- | --- | --- | --- | --- | --- | --- |
| Method | $\mathcal{C} = 25$ | $\mathcal{C} = 50$ | $\mathcal{C} = 75$ | $\mathcal{C} = 25$ | $\mathcal{C} = 50$ | $\mathcal{C} = 75$ | $\mathcal{C} = 25$ | $\mathcal{C} = 50$ | $\mathcal{C} = 75$ |
| PSVAS-F | 2.77 | 4.55 | 5.99 | 4.61 | 6.77 | 8.09 | 4.26 | 6.87 | 9.05 |
| **PSVAS** | **3.19** | **4.83** | **6.34** | **4.69** | **6.94** | **8.12** | **4.95** | **7.56** | **9.51** |
| | Test with Helicopter as Target | | | Test with Plane as Target | | | Test with RB as Target | | |
| Method | $\mathcal{C} = 25$ | $\mathcal{C} = 50$ | $\mathcal{C} = 75$ | $\mathcal{C} = 25$ | $\mathcal{C} = 50$ | $\mathcal{C} = 75$ | $\mathcal{C} = 25$ | $\mathcal{C} = 50$ | $\mathcal{C} = 75$ |
| PSVAS-F | 1.02 | 2.03 | 2.64 | 4.62 | 7.26 | 9.16 | 1.57 | 2.29 | 2.72 |
| **PSVAS** | **1.53** | **2.33** | **2.84** | **5.09** | **7.64** | **9.41** | **1.87** | **2.34** | **2.76** |

Table 24: **ANT** comparisons when trained with *small car* as target on xView in single-query setting.

| | Test with Helicopter as Target | | | Test with SB as Target | | | Test with Building as Target | | |
| --- | --- | --- | --- | --- | --- | --- | --- | --- | --- |
| Method | $\mathcal{C} = 25$ | $\mathcal{C} = 50$ | $\mathcal{C} = 75$ | $\mathcal{C} = 25$ | $\mathcal{C} = 50$ | $\mathcal{C} = 75$ | $\mathcal{C} = 25$ | $\mathcal{C} = 50$ | $\mathcal{C} = 75$ |
| PSVAS-F | 0.55 | 0.86 | 1.24 | 0.66 | 1.12 | 1.34 | 5.88 | 9.45 | 12.23 |
| **PSVAS** | **0.87** | **1.08** | **1.28** | **0.93** | **1.23** | **1.66** | **6.81** | **10.53** | **13.44** |
| | Test with CC as Target | | | Test with SC as Target | | | Test with Helipad as Target | | |
| Method | $\mathcal{C} = 25$ | $\mathcal{C} = 50$ | $\mathcal{C} = 75$ | $\mathcal{C} = 25$ | $\mathcal{C} = 50$ | $\mathcal{C} = 75$ | $\mathcal{C} = 25$ | $\mathcal{C} = 50$ | $\mathcal{C} = 75$ |
| PSVAS-F | 1.45 | 2.30 | 3.01 | 4.84 | 7.56 | 9.65 | 0.82 | 1.20 | 1.46 |
| **PSVAS** | **1.62** | **2.49** | **3.14** | **5.51** | **8.33** | **10.52** | **0.91** | **1.22** | **1.47** |

In Figure 4, the distinct exploration strategy behaviors of PSVAS and PSVAS-F are depicted when both policies are trained with a *large vehicle* as the target and tested with a *ship* as the target. Out of a total of 15 queries, PSVAS-F achieves 6 successful searches, while PSVAS achieves 8 successful searches. Figure 5 illustrates the contrasting exploration strategy behaviors between PSVAS and

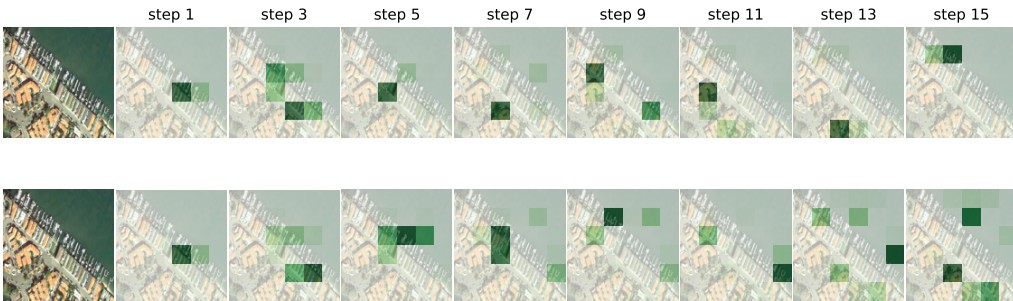

Figure 4: Query sequences, and corresponding heat maps (darker indicates higher probability), obtained using PSVAS-F (top row), PSVAS (bottom row).

PSVAS-F in the case when both the policies are trained with *large vehicle* as the target and test

with *plane* as the target. We observe PSVAS-F yields 9 successful searches, while PSVAS yields 12 successful search out of 15 total query.

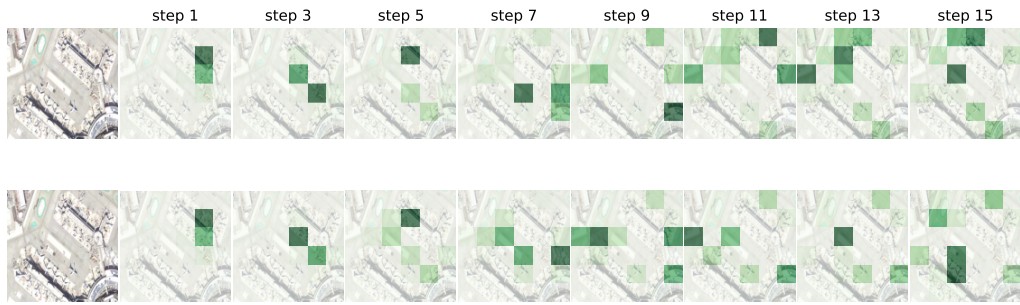

Figure 5: Query sequences, and corresponding heat maps (darker indicates higher probability), obtained using PSVAS-F (top row), PSVAS (bottom row).

Figure 6 illustrates the contrasting exploration strategy behaviors between PSVAS and PSVAS-F in the case when both the policies are trained with *large vehicle* as the target and test with *roundabout* as the target. We observe PSVAS-F yields 5 successful searches, while PSVAS yields 7 successful search out of 15 total query.

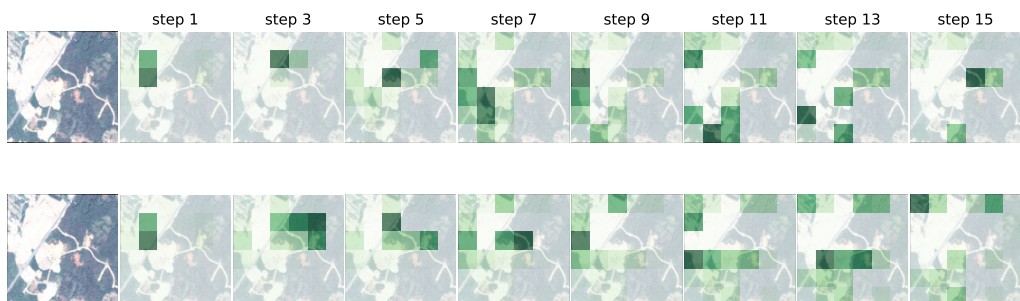

Figure 6: Query sequences, and corresponding heat maps (darker indicates higher probability), obtained using PSVAS-F (top row), PSVAS (bottom row).

## D.2 Effect on MPS-VAS Framework

Next, we examine the influence of inference time adaptation of the task-specific prediction module on the MPS-VAS framework across various target settings. For this purpose, we train a policy using our proposed MPS-VAS approach. But during inference, we freeze both the task-specific prediction module and the task-agnostic search module, which differs from the standard MPS-VAS approach. We refer the resulting policy as MPS-VAS-F. Table 26 presents a comparison of the search performance between MPS-VAS and MPS-VAS-F, considering a grid cell count of $N = 36$, across various target settings. Similarly, in Table 25, we provide corresponding results with a grid cell count of $N = 49$. Across various target settings, we observe a notable enhancement in the performance of MPS-VAS compared to MPS-VAS-F. This finding underscores the significance of adapting the task-specific prediction module during inference after each query, validating its importance on adaptive visual active search. Following Figures demonstrate the divergent exploration strategy behaviors exhibited by MPS-VAS and MPS-VAS-F.

Figure 7 illustrates the contrasting exploration strategy behaviors of MPS-VAS and MPS-VAS-F when both policies are trained with a *large vehicle* as the target and tested with a *plane* as the target. Among a total of 15 queries, MPS-VAS-F achieves 2 successful searches, while MPS-VAS achieves

Table 25: **ANT** comparisons when trained with *small car* as target on xView in single-query setting.

| | Test with Helicopter as Target | | | Test with SB as Target | | | Test with Building as Target | | |
|---|---|---|---|---|---|---|---|---|---|
| Method | $\mathcal{C} = 25$ | $\mathcal{C} = 50$ | $\mathcal{C} = 75$ | $\mathcal{C} = 25$ | $\mathcal{C} = 50$ | $\mathcal{C} = 75$ | $\mathcal{C} = 25$ | $\mathcal{C} = 50$ | $\mathcal{C} = 75$ |
| MPS-VAS-F | 0.54 | 0.89 | 1.22 | 0.64 | 1.14 | 1.37 | 5.97 | 9.31 | 12.04 |
| **MPS-VAS** | **0.92** | **1.13** | **1.38** | **1.07** | **1.67** | **2.10** | **6.83** | **10.59** | **13.64** |
| | Test with CC as Target | | | Test with SC as Target | | | Test with Helipad as Target | | |
| Method | $\mathcal{C} = 25$ | $\mathcal{C} = 50$ | $\mathcal{C} = 75$ | $\mathcal{C} = 25$ | $\mathcal{C} = 50$ | $\mathcal{C} = 75$ | $\mathcal{C} = 25$ | $\mathcal{C} = 50$ | $\mathcal{C} = 75$ |
| MPS-VAS-F | 1.37 | 2.33 | 3.05 | 4.82 | 7.46 | 9.56 | 0.82 | 1.24 | 1.41 |
| **MPS-VAS** | **1.74** | **2.64** | **3.47** | **5.55** | **8.40** | **10.69** | **0.96** | **1.30** | **1.63** |

Table 26: **ANT** comparisons when trained with *LV* as target on DOTA in single-query setting.

| | Test with Ship as Target | | | Test with LV as Target | | | Test with Harbor as Target | | |
|---|---|---|---|---|---|---|---|---|---|
| Method | $\mathcal{C} = 25$ | $\mathcal{C} = 50$ | $\mathcal{C} = 75$ | $\mathcal{C} = 25$ | $\mathcal{C} = 50$ | $\mathcal{C} = 75$ | $\mathcal{C} = 25$ | $\mathcal{C} = 50$ | $\mathcal{C} = 75$ |
| MPS-VAS-F | 2.69 | 4.50 | 5.88 | 4.63 | 6.79 | 8.07 | 4.22 | 6.92 | 9.06 |
| **MPS-VAS** | **3.42** | **5.19** | **6.73** | **4.80** | **7.08** | **8.23** | **5.02** | **8.04** | **9.91** |
| | Test with Helicopter as Target | | | Test with Plane as Target | | | Test with RB as Target | | |
| Method | $\mathcal{C} = 25$ | $\mathcal{C} = 50$ | $\mathcal{C} = 75$ | $\mathcal{C} = 25$ | $\mathcal{C} = 50$ | $\mathcal{C} = 75$ | $\mathcal{C} = 25$ | $\mathcal{C} = 50$ | $\mathcal{C} = 75$ |
| MPS-VAS-F | 1.00 | 2.07 | 2.66 | 4.57 | 7.23 | 9.14 | 1.56 | 2.28 | 2.72 |
| **MPS-VAS** | **1.80** | **2.60** | **3.03** | **5.17** | **7.83** | **10.02** | **1.96** | **2.76** | **3.19** |

4 successful searches. In Figure 8, the distinct exploration strategy behaviors of MPS-VAS and

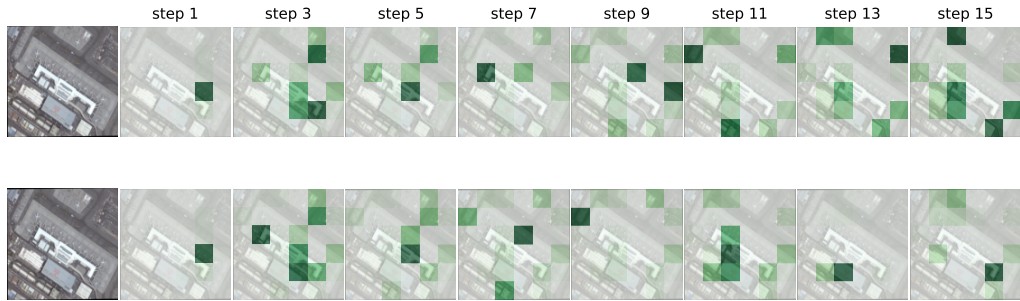

Figure 7: Query sequences, and corresponding heat maps (darker indicates higher probability), obtained using MPS-VAS-F (top row), MPS-VAS (bottom row).

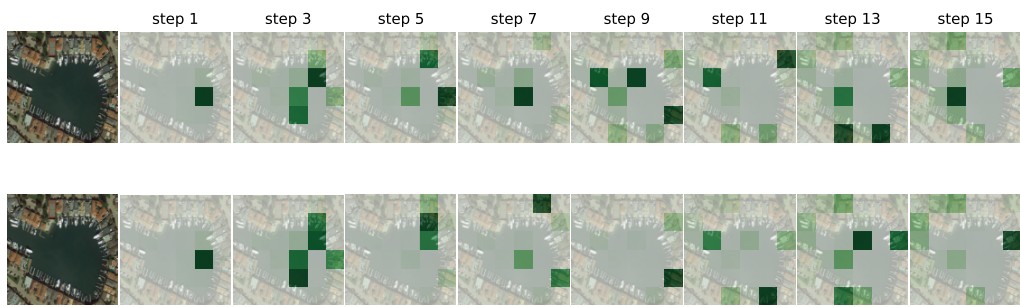

Figure 8: Query sequences, and corresponding heat maps (darker indicates higher probability), obtained using MPS-VAS-F (top row), MPS-VAS (bottom row).

MPS-VAS-F are depicted when both policies are trained with a *large vehicle* as the target and tested with a *ship* as the target. Out of a total of 15 queries, MPS-VAS-F achieves 7 successful searches,

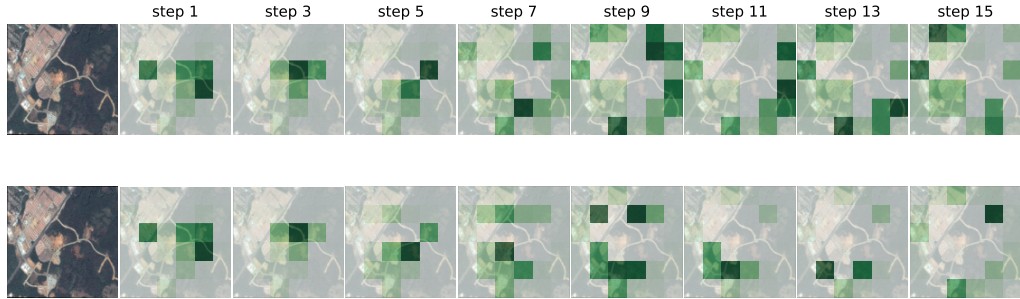

Figure 9: Query sequences, and corresponding heat maps (darker indicates higher probability), obtained using MPS-VAS-F (top row), MPS-VAS (bottom row).

while MPS-VAS achieves 9 successful searches. Figure 9 showcases the contrasting exploration strategy behaviors of MPS-VAS and MPS-VAS-F when both policies are trained with a *large vehicle* as the target and tested with a *roundabout* as the target. Among a total of 15 queries, MPS-VAS-F achieves 6 successful searches, while MPS-VAS achieves 8 successful searches.

# E    More Visualizations of Comparative Exploration Strategies of Different Approaches

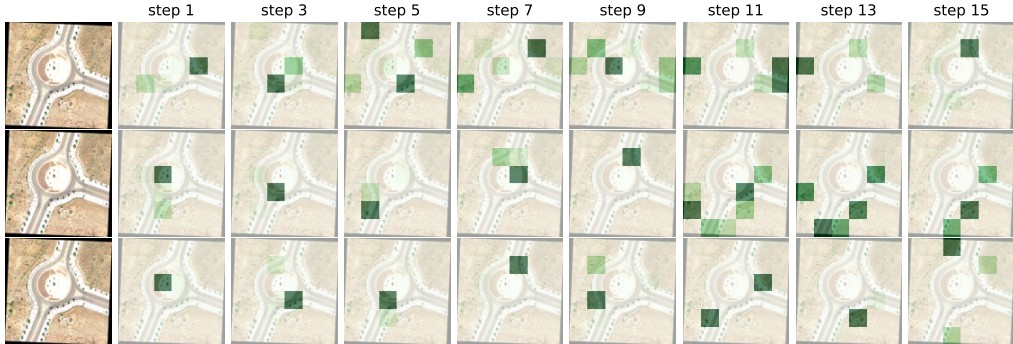

Figure 10: Query sequences, and corresponding heat maps (darker indicates higher probability), obtained using E2EVAS (top row), PSVAS (middle row), and MPS-VAS (bottom row). Note that during the training phase, all these policies are trained with *large vehicle* as the target, while evaluation is conducted using *roundabout* as the target.

The showcased visualizations (10, 11, 12, 13, 14) in all these examples demonstrate the superiority of our PSVAS and MPS-VAS framework compared to the E2EVAS baseline, especially in scenarios where search tasks vary from those employed in policy training.

# F    Analyzing Search Performance Across Multiple Trials

Here, we compare the search performance of E2EVAS, PSVAS, and MPS-VAS across multiple trials. In Figure 15, we present the results when the polices are trained with small car as the target and evaluate the performance under Manhattan distance based query cost $\mathcal{C} = 25$ with the following target classes: *Small Car (SC)*, *Helicopter*, *Sail Boat (SB)*, *Construction Cite (CC)*, *Building*, and *Helipad*. In figure 16, we present similar results with Manhattan distance based query cost budget $\mathcal{C} = 50$. In figure 17, we also present similar results with Manhattan distance based query cost budget $\mathcal{C} = 75$.

In Figure 18, we present the results when the polices are trained with *large vehicle* as the target and evaluate the performance under Manhattan distance based query cost $\mathcal{C} = 25$ with the following target

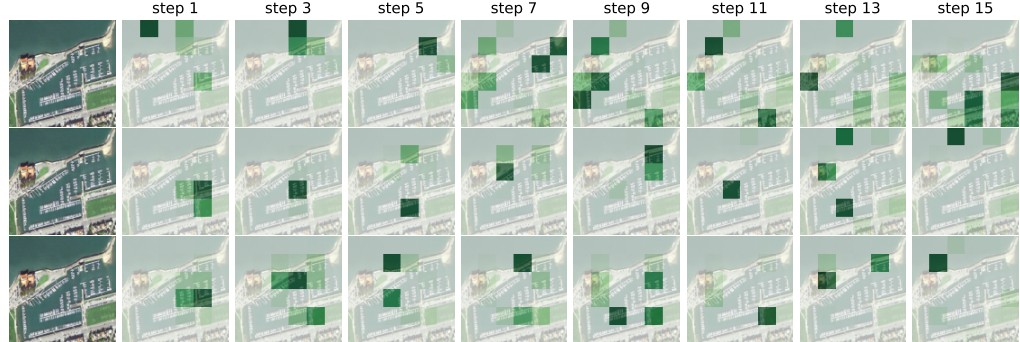

Figure 11: Query sequences, and corresponding heat maps (darker indicates higher probability), obtained using E2EVAS (top row), PSVAS (middle row), and MPS-VAS (bottom row). Note that during the training phase, all these policies are trained with *large vehicle* as the target, while evaluation is conducted using *ship* as the target.

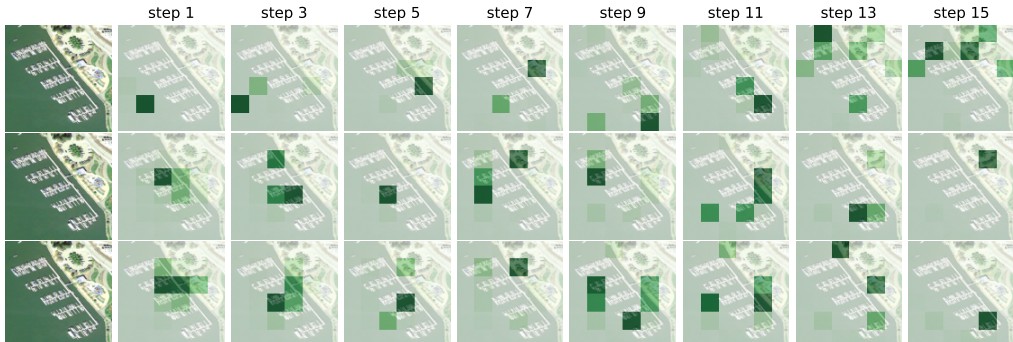

Figure 12: Query sequences, and corresponding heat maps (darker indicates higher probability), obtained using E2EVAS (top row), PSVAS (middle row), and MPS-VAS (bottom row). Note that during the training phase, all these policies are trained with *large vehicle* as the target, while evaluation is conducted using *ship* as the target.

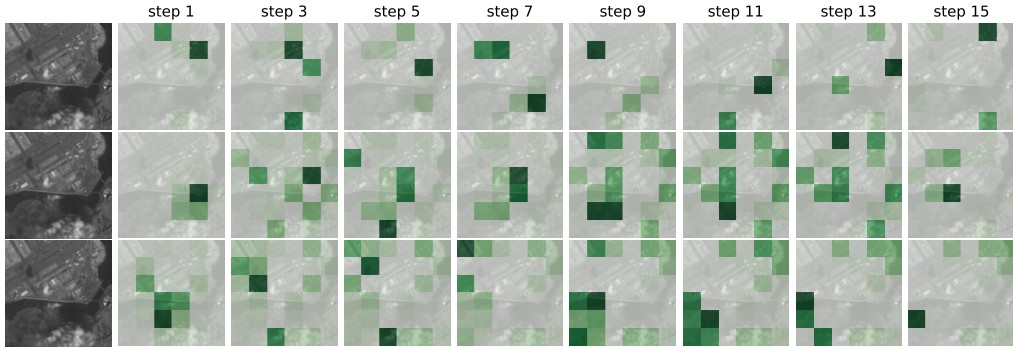

Figure 13: Query sequences, and corresponding heat maps (darker indicates higher probability), obtained using E2EVAS (top row), PSVAS (middle row), and MPS-VAS (bottom row). Note that during the training phase, all these policies are trained with *large vehicle* as the target, while evaluation is conducted using *plane* as the target.

classes: *Ship*, *large vehicle* (LV), *Harbor*, *Helicopter*, *Plane*, and *Roundabout*. In figure 19, we present similar results with Manhattan distance based query cost budget $\mathcal{C} = 50$. In figure 20, we also present similar results with Manhattan distance based query cost budget $\mathcal{C} = 75$.

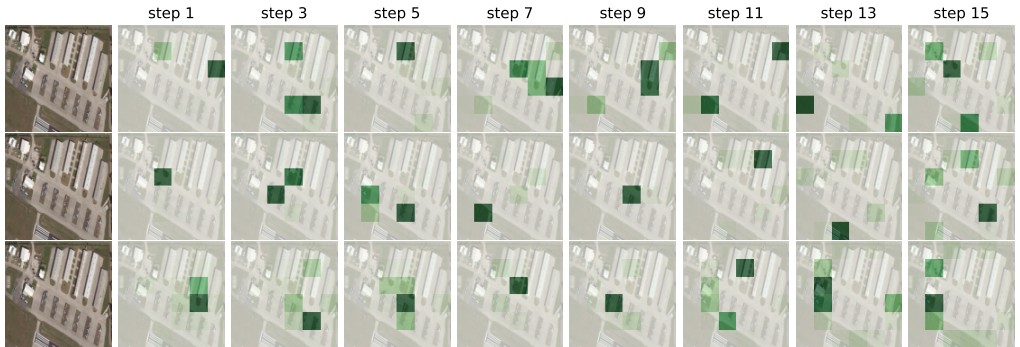

Figure 14: Query sequences, and corresponding heat maps (darker indicates higher probability), obtained using E2EVAS (top row), PSVAS (middle row), and MPS-VAS (bottom row). Note that during the training phase, all these policies are trained with *large vehicle* as the target, while evaluation is conducted using *plane* as the target.

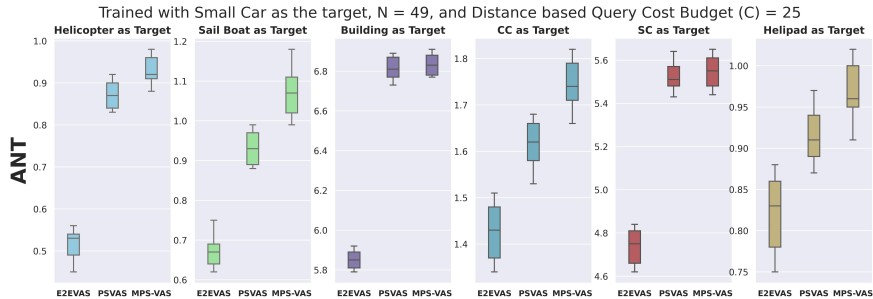

Figure 15: Comparative Search Performance of E2EVAS, PSVAS, MPS-VAS under Distance Based Query Cost ($\mathcal{C} = 25$).

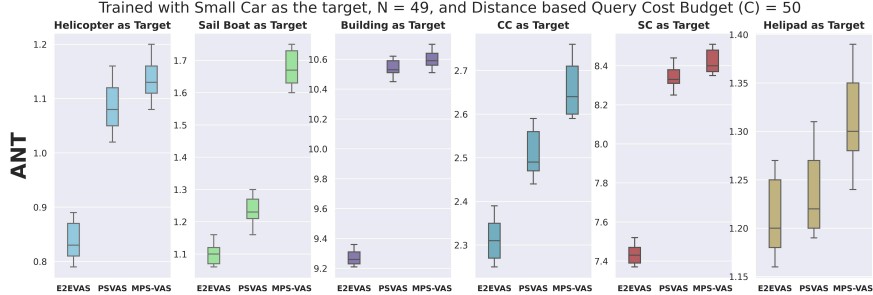

Figure 16: Comparative Search Performance of E2EVAS, PSVAS, MPS-VAS under Distance Based Query Cost ($\mathcal{C} = 50$).

# G   Search Performance Comparisons Across Datasets

Our experimental outcomes indicate that it is possible to apply our method directly across different datasets without requiring any further modifications or hyperparameter tuning. In the following table 27, we demonstrate this by presenting results of training on one dataset for one target class while evaluating on another dataset and for another target class. We use the number of equal sized grid cells $N = 64$ and varying search budgets $\mathcal{C} = \{25, 50, 75\}$.

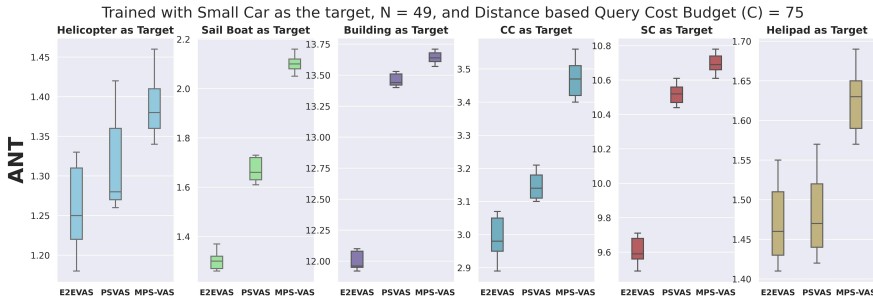

Figure 17: Comparative Search Performance of E2EVAS, PSVAS, MPS-VAS under Distance Based Query Cost ($\mathcal{C}$ = 75).

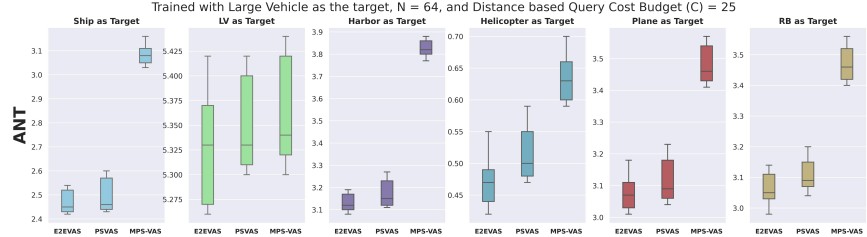

Figure 18: Comparative Search Performance of E2EVAS, PSVAS, MPS-VAS under Distance Based Query Cost ($\mathcal{C}$ = 25).

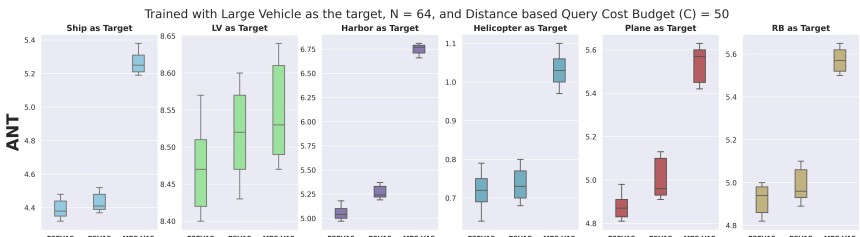

Figure 19: Comparative Search Performance of E2EVAS, PSVAS, MPS-VAS under Distance Based Query Cost ($\mathcal{C}$ = 50).

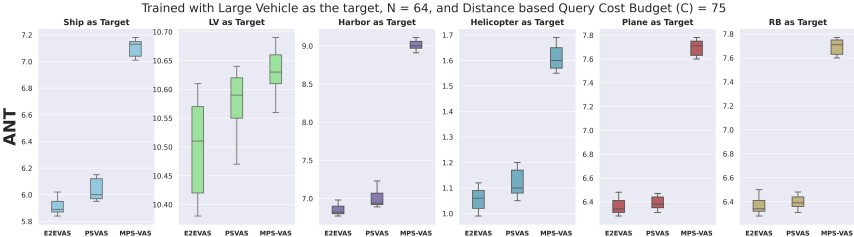

Figure 20: Comparative Search Performance of E2EVAS, PSVAS, MPS-VAS under Distance Based Query Cost ($\mathcal{C}$ = 75).

Table 27: **ANT** comparisons when trained with *large vehicle* on DOTA as target and evaluated with *small car*, *building*, and *sail boat* as target class from xView.

| | Test with Small Car as Target | | | Test with Building as Target | | | Test with Sail Boat as Target | | |
|---|---|---|---|---|---|---|---|---|---|
| $Method$ | $\mathcal{C} = 25$ | $\mathcal{C} = 50$ | $\mathcal{C} = 75$ | $\mathcal{C} = 25$ | $\mathcal{C} = 50$ | $\mathcal{C} = 75$ | $\mathcal{C} = 25$ | $\mathcal{C} = 50$ | $\mathcal{C} = 75$ |
| E2EVAS [6] | 4.22 | 6.73 | 8.07 | 5.12 | 8.24 | 10.50 | 0.48 | 0.56 | 0.92 |
| Online TTA [6] | 4.23 | 6.75 | 8.10 | 5.14 | 8.27 | 10.53 | 0.49 | 0.57 | 0.95 |
| PSVAS | 4.95 | 7.74 | 9.45 | 6.10 | 9.45 | 12.31 | 0.89 | 1.05 | 1.54 |
| MPS-VAS | **5.07** | **7.92** | **9.73** | **6.18** | **9.68** | **12.83** | **1.02** | **1.39** | **1.91** |

