# OpenReview forum: "A Partially-Supervised Reinforcement Learning Framework for Visual Active Search"
_NeurIPS.cc/2023/Conference — NeurIPS 2023 poster_

### Official Review · Reviewer_mjFZ · 2023-07-05

**Soundness:** 3 good
**Presentation:** 3 good
**Contribution:** 3 good
**Rating:** 5
**Confidence:** 3

**Summary:**

This paper introduces a novel Reinforcement Learning (RL) framework for visual active search. In contrast to previous methods, the authors suggest integrating visual supervised signals to enhance performance. Additionally, a meta-learning based approach is proposed to allow the model to adapt to varying test cases. The authors also expand their methodology to support multiple queries. Experimental results demonstrate that the proposed method significantly outperforms both the simple baseline approaches and previous RL-based methods.

**Strengths:**

- The proposed method is both intuitive and technically sound.
- The motivation for test-time adaptation and multi-query adaptation is logically sound and well-justified.
- A variety of datasets and settings were utilized for the experiments, which substantiate the superior performance of the proposed method compared to other approaches.
- The manuscript is well-written and presented, enhancing its clarity and comprehension.

**Weaknesses:**

- Regarding PSVAS, while the concept of simply adding supervised loss is reasonable, it seems more like a straightforward engineering extension to an existing method rather than a novel contribution.
- The authors should conduct more ablation studies to provide readers with a deeper understanding of their method. For instance, an ablation study on hyper-parameters and different modules of the proposed method would be valuable.
- In the proposed method, the supervised loss is merely a binary classification loss that signifies the presence of a target in the query image patch. I wonder if integrating a more fine-grained supervised loss could enhance the overall performance. For example, would adding a detection or segmentation loss be beneficial?
- The paragraph from lines 128 to 140 appears to lack logical consistency. I recommend that the authors restructure this section for improved clarity.

**Questions:**

Comments: Although I don't directly work on this area, I perceive the research problem to be valuable, and the proposed method appears to be sound. However, given my limited expertise, I find it challenging to assess the significance of this paper's technical contributions. Therefore, I've initially rated it as "Borderline Accept". My final rating will depend on feedback from other reviewers, as well as the author's rebuttal.

---

> ### Author Rebuttal · Authors · 2023-08-08
>
> Thank you for the comments and suggestions. We are encouraged to see your appreciation of the writing and presentation style, proposed methodology being intuitive and technically sound, and motivation of TTA and MQA being logically sound and well justified. We respond to all the comments below.
>
>
> > **Q1**: The authors should conduct more ablation studies to provide readers with a deeper understanding of their method. For instance, an ablation study on hyper-parameters and different modules of the proposed method would be valuable.
>
> **A1**: We thank the reviewer for the comment. Indeed, we analyse the importance of the task-specific prediction module in Sections D.1 and D.2 of Supplementary Material by freezing the prediction module parameters during inference time. Here, we additionally analyse the efficacy of the task-specific prediction module by setting **$\lambda$ = 0** while training the policy. We call the resulting policy **USVAS** (Un-Supervised VAS). We observe a significant drop in performance across all settings, demonstrating the importance of the Supervised prediction module in order to learn an effective search policy. Specifically, in the following tables we present the results when the policy is trained with small car on xView as a target, while the performance of the policy is evaluated for the following target classes: Small Car (SC), Helicopter, SailBoat (SB), Construction Site (CS), Building, and Helipad. We evaluate the policy with varying search budgets C ∈ {25, 50, 75} and the number of equal sized grid cells N = 49.
>
> In the following table, we report the test result with **small car** as a target.
>
> | Method  | C = 25  | C = 50  | C = 75 |
> |---|:---:|:---:|:---:|
> |**USVAS** | 4.77 | 7.46 | 9.61 |
> | PSVAS  | 5.51  | 8.33  | 10.52 |
> | MPS-VAS  | **5.55**  | **8.40**  | **10.69**  |
>
> In the following table, we report the test result with **Helicopter** as a target.
>
> | Method  | C = 25  | C = 50  | C = 75 |
> |---|:---:|:---:|:---:|
> |**USVAS** | 0.53 | 0.84 | 1.19 |
> | PSVAS  | 0.87  | 1.08  | 1.28 |
> | MPS-VAS  | **0.92**  | **1.13**  | **1.38**  |
>
> In the following table, we report the test result with **Sail Boat** as a target.
>
> | Method  | C = 25  | C = 50  | C = 75 |
> |---|:---:|:---:|:---:|
> |**USVAS** | 0.64 | 1.08 | 1.27 |
> | PSVAS  | 0.93  | 1.23  | 1.66 |
> | MPS-VAS  | **1.07**  | **1.67**  | **2.10**  |
>
>
> In the following table, we report the test result with **Construction Site** as a target.
>
> | Method  | C = 25  | C = 50  | C = 75 |
> |---|:---:|:---:|:---:|
> |**USVAS** | 1.44 | 2.27 | 2.99 |
> | PSVAS  | 1.62  | 2.49  | 3.14 |
> | MPS-VAS  | **1.74**  | **2.64**  | **3.47**  |
>
> In the following table, we report the test result with **Building** as a target.
> | Method  | C = 25  | C = 50  | C = 75 |
> |---|:---:|:---:|:---:|
> |**USVAS** | 5.86 | 9.37 | 12.05 |
> | PSVAS  | 6.81  | 10.53  | 13.44 |
> | MPS-VAS  | **6.83**  | **10.59**  | **13.64**  |
>
> In the following table, we report the test result with **Helipad** as a target.
>
> | Method  | C = 25  | C = 50  | C = 75 |
> |---|:---:|:---:|:---:|
> |**USVAS** | 0.80 | 1.16 | 1.42 |
> | PSVAS  | 0.91  | 1.22  | 1.47 |
> | MPS-VAS  | **0.96**  | **1.30**  | **1.63**  |
>
> > **Q2**: I wonder if integrating a more fine-grained supervised loss could enhance the overall performance. For example, would adding a detection or segmentation loss be beneficial?
>
> **A2**: This is an intriguing question. Our main objective here is to **find as many target grids** as possible within a pre-specified budget. Consequently, we concentrate on devising an efficient search policy capable of identifying grids containing one or more targets. To achieve an efficient search policy, it becomes crucial to have knowledge about the likely locations of target grids. To address this, we employ a task-specific prediction module trained using BCE loss. However, if we were to shift our focus on a slightly different problem, i.e., Visual Active Target Object Detection, which aims to **precisely identify as many target objects** (along with their exact locations and shapes) as possible within the search budget, using a more fine-grained loss such as detection or segmentation loss would become essential. While this problem of Visual Active Target Object Detection is beyond the scope of our current work, it is a terrific problem for follow up research!
>
> > **Q3**: The paragraph from lines 128 to 140 appears to lack logical consistency. I recommend that the authors restructure this section for improved clarity.
>
> **A3**: Thank you for the suggestion. We will restructure the referenced section in a revised draft.

---

### Official Review · Reviewer_2ixD · 2023-07-06

**Soundness:** 3 good
**Presentation:** 3 good
**Contribution:** 3 good
**Rating:** 6
**Confidence:** 4

**Summary:**

This paper proposes a Partially Supervised Reinforcement Learning to search for the objects of interest in geospatial images. The paper uses a meta-learning approach to learn initialization parameters of the prediction module jointly with a search policy.  A variant of this approach, MPS-VAS, learns  to choose a subset of queries to make in each search iteration. The proposed method outperforms the state-of-the-art and other baselines on xView and DOTA benchmarks.

**Strengths:**

This paper proposes a Partially Supervised Reinforcement Learning to search for the objects of interest in geospatial images. The paper uses a meta-learning approach to learn initialization parameters of the prediction module jointly with a search policy.  A variant of this approach, MPS-VAS, learns  to choose a subset of queries to make in each search iteration. The proposed method outperforms the state-of-the-art and other baselines on xView and DOTA benchmarks. The results show drastically better results than the state-of-the-art model OnlineTTA.

**Weaknesses:**

While the proposed method is promising and novel, it does limit itself to only two benchmarks on satellite images. It would increase the contribution significantly if it was tested on some traditional vision benchmarks. At least, a discussion on it would be useful.

**Questions:**

1. Can we train on a dataset (xView) and generalize to the classes on another dataset (DOTA)?

2. Can we use the same model for image classification task on the fMoW dataset?



**Limitations:**

This paper proposes a Partially Supervised Reinforcement Learning to search for the objects of interest in geospatial images. The paper uses a meta-learning approach to learn initialization parameters of the prediction module jointly with a search policy.  A variant of this approach, MPS-VAS, learns  to choose a subset of queries to make in each search iteration. The proposed method outperforms the state-of-the-art and other baselines on xView and DOTA benchmarks. The results show drastically better results than the state-of-the-art model OnlineTTA. However, the method is not tested on traditional vision benchmarks.

---

> ### Author Rebuttal · Authors · 2023-08-08
>
> We thank the reviewer for the insightful comments and suggestions. We are encouraged to see that you find our proposed method is promising and novel. We respond to all the comments below.
>
> > **Q1**: Can we train on a dataset (xView) and generalize to the classes on another dataset (DOTA)?
>
> **A1**: Absolutely!  Please see the data we provide in response to a similar question by Reviewer 6bym. In a nutshell, we can apply the proposed approach with no modification to train on one dataset with a target object class and then evaluate on a different dataset, and a different target object class. In such cases, our approach also demonstrates superior performance---often by a large margin---compared to the most competitive baselines.
>
> > **Q2**: Can we use the same model for image classification task on the fMoW dataset?
>
> **A2**: Active search is qualitatively distinct from image classification task. The goal of the image classification task is solely to learn to predict well. In active search, in contrast, we aim to learn a search policy that balances exploration (improving our ability to predict where target objects are) and exploitation (actually finding such objects) within a limited budget. The key reason for this balance is that queries are informative about the location of target objects in two ways: 1) geospatial correlations in object locations, and 2) improvement of the quality of the learned predictive model.  In contrast, traditional classification is a sequence of one-shot **prediction** tasks, rather than **search** tasks.  Consequently, traditional vision benchmarks do not provide an appropriate evaluation framework for our problem.
>
> > **Q3**: It would increase the contribution significantly if it was tested on some traditional vision benchmarks. At least, a discussion on it would be useful.
>
> **A3**: As we mention in our response to the question above, visual active search (VAS) is a qualitatively different problem than traditional vision tasks, and therefore typical vision benchmarks are not appropriate means for evaluating VAS approaches, as we are interested primarily in the quality of visual search (trading off exploration and exploitation) rather than the quality of visual prediction.  We agree that this warrants further discussion and clarification, which we will add in the revision.

---

### Official Review · Reviewer_6bym · 2023-07-08

**Soundness:** 3 good
**Presentation:** 3 good
**Contribution:** 3 good
**Rating:** 6
**Confidence:** 4

**Summary:**

The paper proposes a partially supervised reinforcement learning framework (PSVAS) for visual active search tasks (VAS). Specifically, it decompose previous end-to-end deep RL method for VAS into a task-specific prediction module and a task-agnostic search module. In such way, the prediction module can be supervised with observed ground truth during training or inference time. Further, a meta-learning setting is added to adapt to different tasks quickly. Results on xView and DOTA datasets show the proposed methods and variants outperform previous baselines.

**Strengths:**

The idea to utilize the target-location labels as additional supervision signal besides RL loss is easy to follow and makes sense. The method shows much better adaptation capability when applying to search target that is not trained on, demonstrating the advantages of decomposing the problem into prediction and searching separately. The meta-learning variants and multi-query variants further enhanced the performance. Overall the method is techniquely sound and empirically proved.

**Weaknesses:**


How different balancing factor lambda between RL and BCE loss would affect the performance is worth to investigate in ablation study section.


**Questions:**


Can authors address the weaknesses part.


**Limitations:**

Whether it is possible to apply such method across different datasets is not discusses.

---

> ### Author Rebuttal · Authors · 2023-08-08
>
> Thank you for the thoughtful comments and suggestions. We are inspired to see that you found our overall method is techniqually sound and empirically proved. We respond to all the comments below.
>
> > **Q1**: How different balancing factor lambda between RL and BCE loss would affect the performance is worth to investigate in ablation study section.
>
> **A1**: We performed experiments with different choices of $\lambda$ and found $\lambda$ = 0.1 to be the best choice across all different experimental setup. For comparison, here we report the results in the case when we train the policy with different values of $\lambda$ using small car as a target class and test the policy with small car, building, and sail boat as target on xView. We evaluate the policy with varying search budgets C ∈ {25, 50, 75} and the number of equal sized grid cells N = 49.
>
> In the following table, we provide the result for the **PSVAS** framework with **small car** as target.
>
> | $\lambda$  | C = 25  | C = 50  | C = 75 |
> |---|:---:|:---:|:---:|
> |0.001| 4.96 | 7.75 | 9.74 |
> | 0.01  | 5.02  | 7.87  | 9.96  |
> | 0.1  | **5.51**  | **8.33**  | **10.52**  |
> | 1.0  | 5.10  | 7.98  | 10.04  |
>
> In the following table, we provide the result for **MPS-VAS** framework with **small car** as target.
>
> | $\lambda$  | C = 25  | C = 50  | C = 75 |
> |---|:---:|:---:|:---:|
> |0.001| 4.99 | 7.82 | 9.90 |
> | 0.01  | 5.06  | 7.93  | 10.03  |
> | 0.1  | **5.55**  | **8.40**  | **10.69**  |
> | 1.0  | 5.12  | 8.01  | 10.12  |
>
> In the following table, we provide the result for **PSVAS** framework with **building** as target.
>
> | $\lambda$  | C = 25  | C = 50  | C = 75 |
> |---|:---:|:---:|:---:|
> |0.001| 6.08 | 9.64 | 12.35 |
> | 0.01  | 6.37  | 9.95  | 12.77  |
> | 0.1  | **6.81**  | **10.53**  | **13.44**  |
> | 1.0  | 6.39  | 10.16  | 12.81  |
>
> In the following table, we provide the result for **MPS-VAS** framework with **building** as target.
>
> | $\lambda$  | C = 25  | C = 50  | C = 75 |
> |---|:---:|:---:|:---:|
> |0.001| 6.15 | 9.74 | 12.44 |
> | 0.01  | 6.41  | 10.09  | 12.89  |
> | 0.1  | **6.83**  | **10.59**  | **13.64**  |
> | 1.0  | 6.46  | 10.21  | 12.96  |
>
> In the following table, we provide the result for **PSVAS** framework with **sail boat** as target.
>
> | $\lambda$  | C = 25  | C = 50  | C = 75 |
> |---|:---:|:---:|:---:|
> |0.001| 0.74 | 1.12 | 1.43 |
> | 0.01  | 0.88  | 1.19  | 1.54  |
> | 0.1  | **0.93**  | **1.23**  | **1.66**  |
> | 1.0  | 0.89  | 1.20  | 1.59  |
>
> In the following table, we provide the result for **MPS-VAS** framework with **sail boat** as target.
>
> | $\lambda$  | C = 25  | C = 50  | C = 75 |
> |---|:---:|:---:|:---:|
> |0.001| 0.83 | 1.22 | 1.53 |
> | 0.01  | 0.98  | 1.46  | 1.87  |
> | 0.1  | **1.07**  | **1.67**  | **2.10**  |
> | 1.0  | 1.01  | 1.52  | 1.90  |
>
> Our empirical findings across all the experimental settings are quite consistent, and justify the choice of $\lambda=0.1$.
>
> > **Q2**: Whether it is possible to apply such method across different datasets is not discusses.
>
> **A2**: We can apply our method directly across different datasets without requiring any further modifications or hyperparameter tuning. In the following tables, we demonstrate this by presenting results of training on one dataset for one target class while evaluating on another dataset and for another target class. We use the number of equal sized grid cells N = 64 and varying search budgets C = {25,50,75}.
>
> In the following table, we report the results when we **train** the policy with **large vehicle on DOTA** as target and **evaluate** with **small car on xView** as target.
>
> | Method  | C = 25  | C = 50  | C = 75 |
> |---|:---:|:---:|:---:|
> |E2EVAS | 4.22 | 6.73 | 8.07 |
> |OnlineTTA | 4.23 | 6.75 | 8.10 |
> | PSVAS  | 4.95  | 7.74  | 9.45 |
> | MPS-VAS  | **5.07**  | **7.92**  | **9.73**  |
>
> In the next table, we report the results when we **train** the policy with **large vehicle on DOTA** as target and **evaluate** with **building on xView** as target.
>
> | Method  | C = 25  | C = 50  | C = 75 |
> |---|:---:|:---:|:---:|
> |E2EVAS | 5.12 | 8.24 | 10.50 |
> |OnlineTTA | 5.14 | 8.27 | 10.53 |
> | PSVAS  | 6.10  | 9.45  | 12.31 |
> | MPS-VAS  | **6.18**  | **9.68**  | **12.83**  |
>
> In the next table, we report the result when we **train** the policy with **large vehicle on DOTA** as target and **evaluate** with **sail boat on xView** as target.
>
> | Method  | C = 25  | C = 50  | C = 75 |
> |---|:---:|:---:|:---:|
> |E2EVAS | 0.48 | 0.56 | 0.92 |
> |OnlineTTA | 0.49 | 0.57 | 0.95 |
> | PSVAS  | 0.89  | 1.05  | 1.54 |
> | MPS-VAS  | **1.02**  | **1.39**  | **1.91**  |
>
> Our experimental findings suggest that our proposed PSVAS and MPS-VAS framework significantly improves ANT compared to the most competitive baselines even in the case when the training and evaluation datasets and target objects differ.

---

> > ### Comment · Reviewer_6bym · 2023-08-19
> > **Thanks for the author response**
> >
> > Thank authors for the detailed response regarding the balance weight and generation to other datasets. The provided results have resolved my concerns.

---

### Official Review · Reviewer_yVuL · 2023-07-16

**Soundness:** 3 good
**Presentation:** 2 fair
**Contribution:** 3 good
**Rating:** 5
**Confidence:** 2

**Summary:**

The paper addresses visual active search problem for geospatial exploration. For an effective adaptation of search policies to out-of-distribution tasks, the paper proposes partially supervised reinforcement learning framework. A meta-learning approach is also proposed that focuses on jointly learning a policy and initialization parameters for the supervised prediction module. Experiments are presented on two datasets.

**Strengths:**

+ Proposed partially supervised reinforcement learning framework allows effective adaptation of search policies to out-of-distribution tasks.
+ Proposed meta-learning framework focuses on effective learning of supervised prediction module
+ Experiments show promising results

**Weaknesses:**

-- Not sure of the reproducibility of this work. Also, no code is provided in the supplementary.
-- If there is fully labeled training task, why imitation learning (IL) is not adopted? IL and RL loss can be used together for the task.
-- It will be good to provide the formulation of L_BCE and L_RL loss in the paper?

**Questions:**

Please see weaknesses.

---

> ### Author Rebuttal · Authors · 2023-08-08
>
> Thank you for your comments. We are encouraged to see that you appreciated our contribution. We respond to all the comments below.
>
> > **Q1**:  Not sure of the reproducibility of this work. Also, no code is provided in the supplementary.
>
> **A1**: With anonymity being a primary concern, we have chosen not to release the code at this point. Nevertheless, we are committed to open-sourcing the GitHub link upon acceptance, inclusive of all model weights, to facilitate the reproduction of all our experimental results.
>
> > **Q2**: why imitation learning (IL) is not adopted? IL and RL loss can be used together for the task.
>
> **A2**: We wish to clarify that for imitation learning to be applied, we would need examples of search demonstrations by people, which we do not have.  Rather, our labels provide information about object locations across grid cells.  The reason this is not useful for an imitation learning paradigm is that the search task is fundmentally about handling unknown object locations, leveraging previous observations about these, and trading off exploration and exploitation.  Consequently, in our setting the only way we can make use of imitation learning is to learn to greedily choose a location most likely to contain an object.  And, in fact, such a greedy approach is one of our baselines (we refer to it as greedy classification (GC)), and performs rather poorly (e.g., in Table 1, it is one of the weakest baselines).
>
> > **Q3**: It will be good to provide the formulation of L_BCE and L_RL loss in the paper?
>
> **A3**: Thank you for the suggestion. We thought the formulation of binary cross-entropy loss ($L_{BCE}$) and REINFORCE loss ($L_{RL}$) is well-known and not necessary to include. But, following your suggestion we will include this in the revision.

---

> > ### Author Response · Authors · 2023-08-16
> >
> > Dear Reviewer yVuL,
> >
> > With the rebuttal/discussion period nearing a close, please let us know if you have further questions/clarifications.  In particular, please let us know if you have any specific lingering concerns about reproducibility. As we mentioned in the rebuttal, we intend to release all code (including both for training, inference, and evaluation), learned models, and data, along with detailed documentation on github that enable easy reproduction of all the results in the paper, if it is accepted.  Since github is not anonymous, and the supplement has a size limitation, we were unable to provide these materials as a part of the original submission (for example, the learned models themselves are quite large).

---

### Author Rebuttal · Authors · 2023-08-08

We thank all the reviewers for their constructive and insightful feedback. We are delighted to see that our work is found novel and technically solid by all the reviewer. We’ve addressed all reviewer concerns to the best of our effort by adding detailed clarifications with supporting experimental results, and will incorporate the feedback in the final version.

---

### Comment · Area_Chair_e7WP · 2023-08-20
**Thanks for your response -- AC Comment**

Dear authors,

Thanks for providing your responses to the reviewers' comments. They are comprehensive and answer many of the concerns raised during the initial review phase. While there is still time in the discussion phase, the reviewers can engage for further clarifications.

-- Your AC

---

### Decision · Program_Chairs · 2023-09-21

**Decision:**

Accept (poster)

**Comment:**

The work received generally positive reviews. Concerns were raised about generalization, ablations, and baselines during the review period. The authors provided several responses that attempt to clarify these questions and generally address most of the concerns. The authors are encouraged to incorporate the information from the rebuttal into the final version for completeness.